# Microglial calcium signaling is attuned to neuronal activity in awake mice

**Anthony D Umpierre[1], Lauren L Bystrom[1], Yanlu Ying[1], Yong U Liu[1], Gregory Worrell[1], Long-Jun Wu[1,2,3]***

[1]Department of Neurology, Mayo Clinic, Rochester, United States; [2]Department of Neuroscience, Mayo Clinic, Jacksonville, United States; [3]Department of Immunology, Mayo Clinic, Rochester, United States

**Abstract** Microglial calcium signaling underlies a number of key physiological and pathological processes in situ, but has not been studied in vivo in awake mice. Using multiple GCaMP6 variants targeted to microglia, we assessed how microglial calcium signaling responds to alterations in neuronal activity across a wide range. We find that only a small subset of microglial somata and processes exhibited spontaneous calcium transients in a chronic window preparation. However, hyperactive shifts in neuronal activity (kainate status epilepticus and CaMKIIa Gq DREADD activation) triggered increased microglial process calcium signaling, often concomitant with process extension. Additionally, hypoactive shifts in neuronal activity (isoflurane anesthesia and CaMKIIa Gi DREADD activation) also increased microglial process calcium signaling. Under hypoactive neuronal conditions, microglia also exhibited process extension and outgrowth with greater calcium signaling. Our work reveals that microglia have highly distinct microdomain signaling, and that processes specifically respond to bi-directional shifts in neuronal activity through increased calcium signaling.

***For correspondence:**
Wu.LongJun@mayo.edu

**Competing interests:** The authors declare that no competing interests exist.

## Introduction

Microglia are resident immune cells of the CNS, which are specialized to respond to both immunological and neuronal stimuli. Early work in situ suggested that calcium activity was critical for key microglial functions such as motility, cytokine release, and receptor trafficking/diffusion (*Färber and Kettenmann, 2006*; *Korvers et al., 2016*; *Toulme and Khakh, 2012*). However, more recent in vivo studies have demonstrated that spontaneous calcium signaling is nearly absent in microglia (*Brawek et al., 2017*; *Eichhoff et al., 2011*; *Pozner et al., 2015*). Such observations downplay the potential utility of calcium signaling for microglial function.

Over the past two decades of astrocyte calcium research, it has become clear that microdomains (soma, major branches, and minor branches) have exquisitely different calcium signaling properties, with the smallest sub-compartments of the cell having the most abundant calcium signaling (*Srinivasan et al., 2015*). Microglial processes are well known to dynamically extend and retract as they survey the brain microenvironment (*Eyo and Wu, 2019*; *Nimmerjahn et al., 2005*; *Wake et al., 2009*). To date, microglial processes have not been extensively evaluated for their calcium activity in vivo, nor has any assessment been performed in the awake animal. Herein, we evaluated microglial soma and process calcium signaling in awake mice using two-photon microscopy.

One of the key questions in microglial research is the extent to which microglia functionally respond to neuronal activity in the mature brain. Testing a high signal-to-noise GCaMP variant (GCaMP6s) and a membrane-tethered GCaMP variant (Lck-GCaMP6f), we find that microglia have very infrequent calcium activity in their processes under basal conditions. However, microglial process calcium signaling is attuned to neuronal activity changes, with both neuronal hypoactivity and hyperactivity triggering increases in microglial process calcium signals. By contrast, microglial

somatic calcium changes may represent a disease signature as somatic calcium changes were only evident in longitudinal epilepsy development. Our work reveals that microglia have highly dynamic process calcium activity during network activity shifts, most closely associated with processes undergoing extension or outgrowth.

## Results

### Spontaneous microglial calcium activity in Lck-GCaMP6f and cytosolic GCaMP6s mice

Microglial calcium activity has been frequently described in situ, with limited studies performed in vivo. We first assessed microglial calcium activity in acute window preparations (24 hr after surgery), testing the utility of two sensors: GCaMP6s and Lck-GCaMP6f (*Madisen et al., 2015*; *Srinivasan et al., 2016*). The GCaMP6s sensor was chosen for its ideal signal-to-noise ratio, while the Lck-GCaMP6f sensor was chosen to detect potential fast calcium transients, or transients highly localized to fine processes, as demonstrated in astrocytes (*Srinivasan et al., 2016*). Both sensors were targeted to microglia using the CX3CR1[CreER-IRES-eYFP] mouse line. It is important to note that the low-level, but constitutive eYFP signal from this mouse was detected alongside the dynamic GCaMP6 calcium signal. Spontaneous calcium activity was studied in the awake, head-restrained animal (*Figure 1A*), using two-photon microscopy to assess microglia in layer I and II/III of somatosensory cortex. Microglial calcium activity was present in both the Lck-GCaMP6f mouse and the cytosolic GCaMP6s mouse (*Figure 1B and C*). However, the higher fluorescence amplitude and slower decay kinetics in the GCaMP6s animal made this fluorophore ideal for studying calcium activity in thin microglial processes. On average, GCaMP6s reported calcium transients with a 50% greater amplitude and a 60% increase in signal area (*Figure 1D*). For these reasons, we could detect calcium events in a greater percentage of microdomains in the GCaMP6s animal (*Figure 1C*). Therefore, we adopted GCaMP6s mice for our studies of microglial calcium activity. It should be mentioned, however, that Lck-GCaMP6f mice do report unique phenomenon not observed in the GCaMP6s mouse. Specifically, small amplitude, widespread, and temporally coordinated calcium events can be observed in the parenchyma of Lck-GCaMP6f mice using a grid-based analysis (*Figure 1E*).

Notably, as immune cells, microglia respond to a variety of stimuli, including injury and inflammation. We next assessed whether studying microglial calcium activity soon after surgery in the acute window preparation (>24 hr) resulted in similar calcium activity patterns to a chronic window animal (4 weeks after surgery; *Figure 2A*). The morphology of cortical microglia was highly similar in acute and chronic window studies (Sholl analysis, *Figure 2B*). Despite the similarities in morphology, microglial process calcium activity was significantly greater in acute window preparations, while somatic calcium activity levels were indistinguishable (*Figure 2C–G*). Approximately one-third of microglia somata exhibited calcium activity in layer I (55–75 μm depth) and layer II/III (150–170 μm depth) in both the acute and chronic window preparations (*Figure 2E*). In the acute window preparation, most microglia were either completely inactive (24% of cells surveyed) or had spontaneous activity in under half of their processes (40% of cells surveyed; *Figure 2—figure supplement 1*). However, one month after surgical recovery, the proportion of active processes dropped further for both layer I microglia (56% to 35%), and layer II/III microglia (54% to 41%, *Figure 2E*). Additionally, highly active processes sporadically observed across microglia in the acute window preparation were not maintained in the chronic window animal (*Figure 2G*). Notably, in the chronic window animal, baseline recordings from the same region were stable over 3 successive days of recording (*Figure 2H*). For these reasons, we performed all other studies in chronic window mice to avoid any potential confounds associated with post-surgery microglial activation (*Xu et al., 2007*).

### Isoflurane anesthesia increases microglial process extension and calcium activity

To date, it is unclear whether neuronal activity influences microglial calcium signaling. We first assessed whether general anesthesia impacts microglial calcium signaling. After recording baseline calcium activity in the awake animal, we introduced isoflurane anesthesia via nose cone (*Figure 3A*). Importantly, the isoflurane concentration chosen for maintenance (1.5–2% in oxygen) is intended to

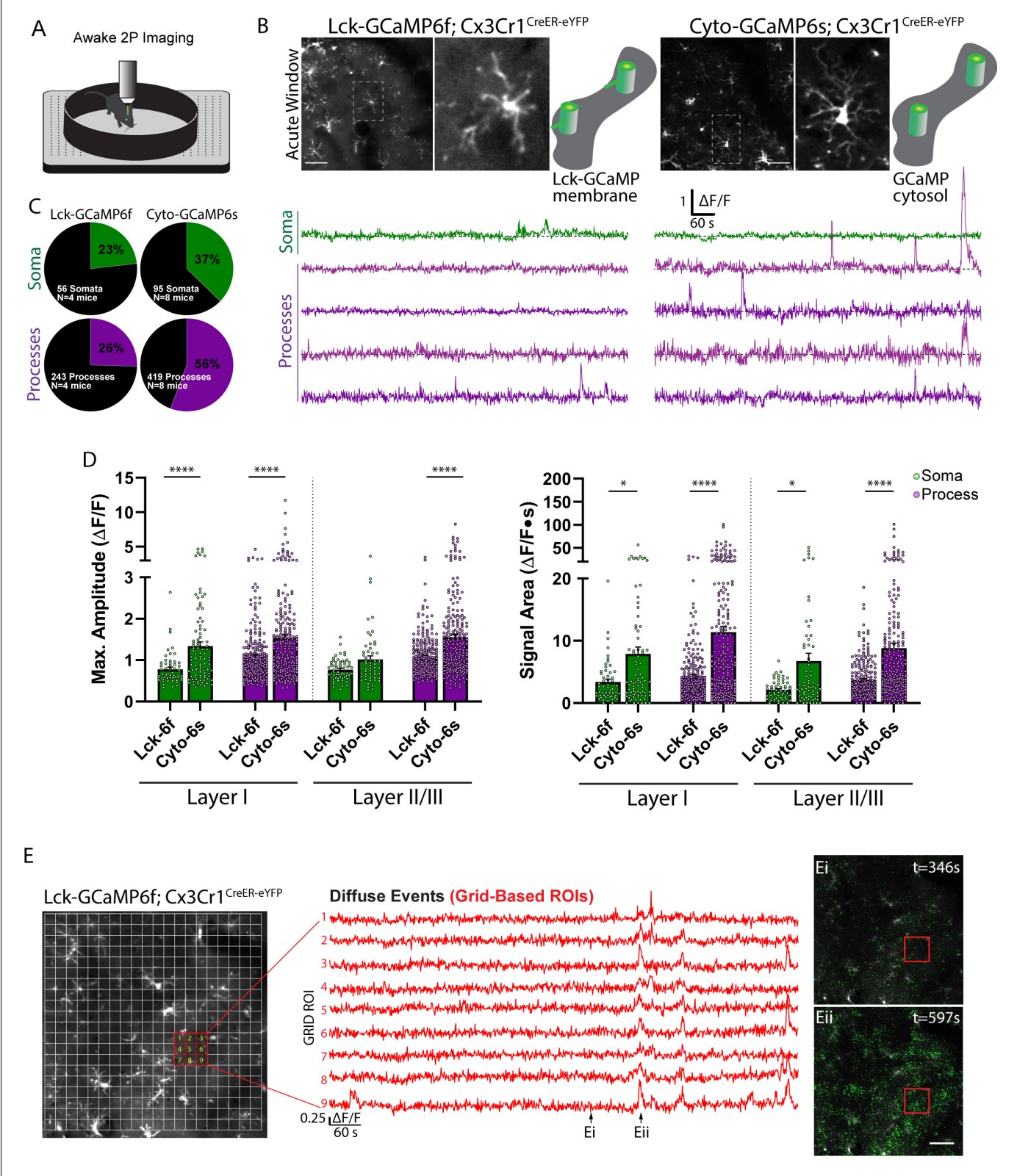

**Figure 1.** Spontaneous microglial calcium activity reported in the Lck-GCaMP6f and cytosolic GCaMP6s mouse. (**A**) Imaging setup: in vivo two-photon imaging of microglial calcium activity in the awake, head-restrained animal. (**B**) Average intensity projection images and microglial ΔF/F calcium traces reported by Lck-GCaMP6f or cytosolic-GCaMP6s mice with a magnified cell. ΔF/F traces are from the soma and processes of a representative cell using manual segmentation. (**C**) Percentage of active microdomains reported by each sensor. (**D**) Maximum amplitude and signal area reported by each

*Figure 1 continued on next page*

Figure 1 continued

sensor (bar: mean ± SEM; dots: individual microdomains; 1-Way ANOVA with Sidak's post-hoc comparison). (E) The Lck-GCaMP6f mouse reports microglial calcium activity that is widespread, coordinated, and low-level throughout the parenchyma (Eii), compared to periods of relative inactivity (Ei). Scale bar: 50 μm (B and E). *p<0.05, ****p<0.0001. N = 4 Lck-GCaMP6f mice, N = 8 Cyto-GCaMP6s mice.

The online version of this article includes the following source data for figure 1:

**Source data 1.** Microglial spontaneous calcium.

mimic strong sensory blockade during surgery and is higher than the concentration typically used during anesthetized imaging (0.5–1.5%). Our studies of neuronal activity indicate that 1.5–2% isoflurane maintenance can reduce excitatory calcium signaling by 91 ± 0.11% in the somatosensory cortex (AAV.CaMKIIa.GCaMP6s.WPRE transfected mice, layer II/III of somatosensory cortex; *Figure 3B–D*). In response to strongly reduced neuronal activity, microglia demonstrate an increase in process calcium activity (*Figure 3E–G* and *Figure 3—Video 1*). Increased microglial process calcium activity is first detected 6 min after anesthesia (*Figure 3H*) and reaches peak levels approximately 10–20 min after isoflurane induction (*Figure 3G*). At their peak, microglial processes display a 290 ± 29% increase in average ΔF/F calcium signal area (*Figure 3G*). Despite strong increases in process calcium activity, somatic calcium signaling remained unaffected in microglia under general anesthesia, with no changes in the proportion or signal area of somatic events (*Figure 3G*). These results provide early evidence that microglial compartments could have distinct calcium-associated functions.

In accordance with our previous studies in the awake animal, general anesthesia increases microglial process outgrowth (*Liu et al., 2019*). At the isoflurane concentration used, we observed a gradual increase in total process area co-occurring with increased process calcium activity (*Figure 3I* and *Figure 3—figure supplement 1*). In situ, microglial process motility has been strongly associated with calcium activity (*Langfelder et al., 2015*; *Nolte et al., 1996*). In vivo, we observe that microglia have a unique morphological response to isoflurane, characterized by extension and outgrowth of sub-branchlets from main processes (*Figure 3J* and *Figure 3—figure supplement 1*). These areas of new process extension/outgrowth exhibit the largest calcium signals and the highest association with calcium activity (*Figure 3J* and *Figure 3—figure supplement 1*). On the other hand, processes with little-to-no motility (termed 'stable' processes) had significantly less associated calcium activity and signal magnitude. Processes that retracted during isoflurane exposure also exhibited much lower associated calcium activity and signal magnitude (*Figure 3I*). Our observations suggest that calcium activity increases under isoflurane are most directly associated with process extension and sub-branchlet outgrowth.

## Kainate administration leads to acute and longitudinal increases in microglial calcium signaling

We additionally used kainate, an excitatory receptor agonist and chemoconvulsant, to determine if strong increases in neuronal activity can alter microglial calcium signaling. In a first cohort of chronic window mice, we recorded microglial calcium activity at baseline and soon after kainate-induced generalized seizures (i.p. kainate model; *Umpierre et al., 2016*) in awake mice (*Figure 4A*). A first generalized seizure began between 45 and 75 min after kainate administration. Seizure generalization involves activation of cortical neuronal populations. In neuronal studies (AAV.CaMKIIa. GCaMP6s.WPRE transfected mice), kainate increased excitatory calcium activity both soon after administration (15–30 min) and after the first observed seizure (55-fold increase in signal area; *Figure 4B and C*). After the first observed seizure, microglial process calcium activity was increased by 360 ± 26%, with a greater proportion of processes exhibiting calcium activity (32% to 73%; *Figure 4D–F*). Despite strong increases in process calcium signaling, microglial somatic signaling did not increase significantly as measured through average signal area; however, there was an increase in the proportion of somata exhibiting calcium activity (29% to 44%; *Figure 4E and F*). During kainate-induced seizures, the predominant motility behavior was process extension, with process area increasing across the parenchyma (*Figure 4G*). These findings demonstrate that both an approach to rapidly decrease neuronal activity (isoflurane) and an approach to rapidly increase neuronal activity (kainate) similarly result in large-scale increases in microglial process calcium activity. In addition,

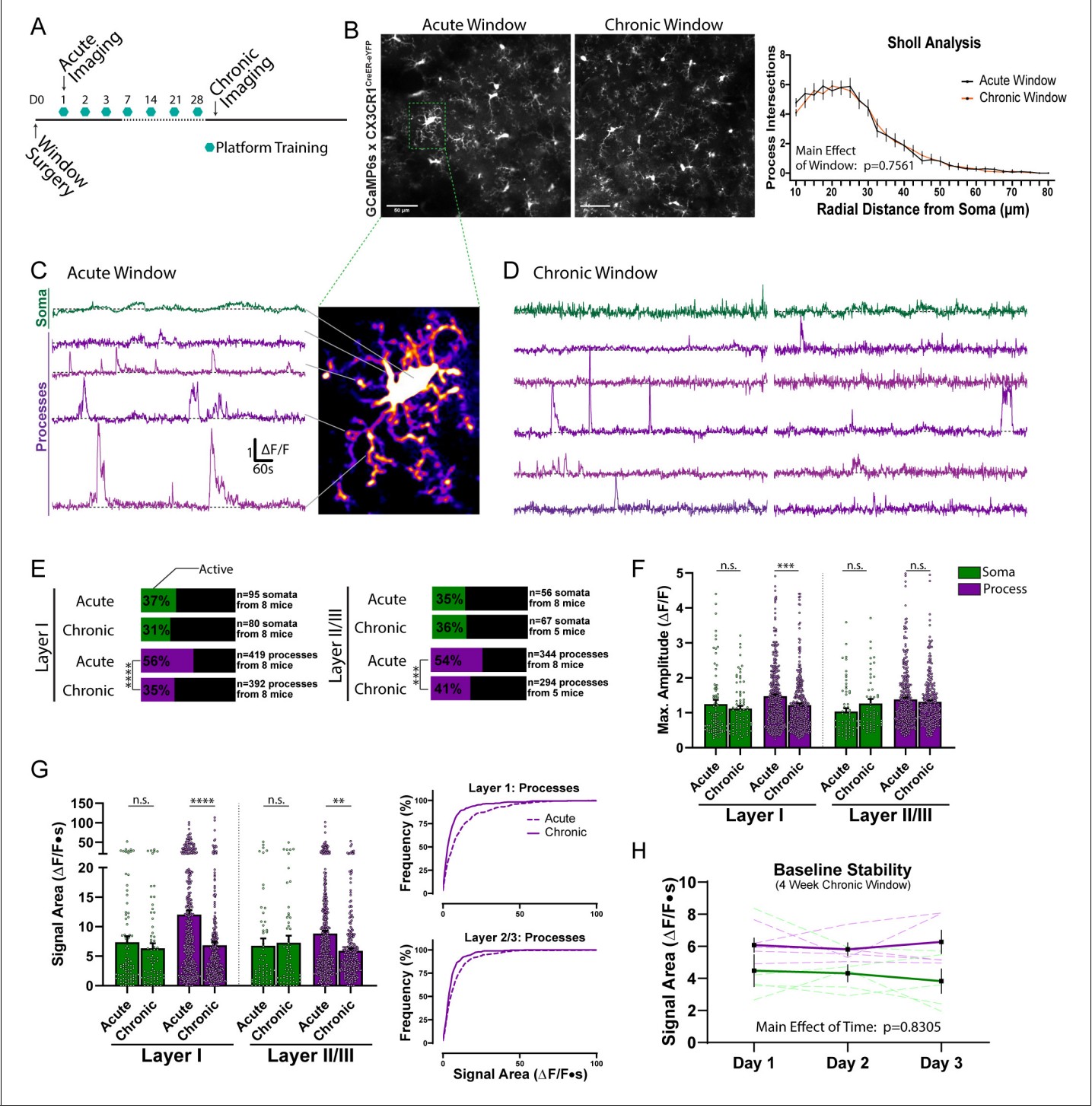

**Figure 2.** Spontaneous microglial calcium activity differs between acute and chronic window preparations. (**A**) Timeline of acute and chronic window imaging and animal training. (**B**) eYFP average intensity images of microglia morphology in acute and chronic window preparations. Sholl analysis of microglial morphology (n = 50 microglia per group; two-way ANOVA). (**C–D**) Microglial ΔF/F calcium traces from the soma and processes of a representative cell in an acute window animal (**C**), or two representative cells in a chronic window animal (**D**), using manual segmentation. (**E**) Percent of active microdomains (Fisher's exact test). (**F**) The amplitude recorded from microdomains (one-way ANOVA with Sidak's post-hoc comparison). (**G**) The calcium signal area recorded from microdomains (one-way ANOVA with Sidak's post-hoc comparison). Cumulative distribution curves for Layer I and Layer II/III process calcium signal areas. (**H**) Microglia soma and process calcium signal areas recorded over 3 successive days in chronic window animals. Scale bars: 50 µm (**B**). Grouped data represent the mean ± SEM; dots represent microdomains (**F** and **G**); dashed lines represent individual animals (**H**). The number of animals and microdomains surveyed is described in (**E**). **p<0.01, ***p<0.001, ****p<0.0001.

*Figure 2 continued on next page*

*Figure 2 continued*

The online version of this article includes the following source data and figure supplement(s) for figure 2:

**Source data 1.** Acute and chronic window spontaneous microglial calcium.
**Figure supplement 1.** Variability in spontaneous microglial calcium activity in acute window preparations.

both approaches resulted in increased microglial process territory and were insufficient to alter microglial somatic calcium activity.

The prolonged seizure state induced by kainate, called status epilepticus, can also serve as a precipitating insult for the later acquisition of epilepsy (*Puttachary et al., 2015*; *Tse et al., 2014*; *Umpierre et al., 2016*). In a second cohort of chronic window mice, we recorded baseline microglial calcium activity and then visually monitored status epilepticus severity after kainate administration (*Figure 5A*). Mice with more severe status epilepticus ($\geq$8 generalized seizures) were longitudinally studied over a 14-day period for changes in microglial calcium signaling. For up to 7 days after kainate status epilepticus, layer I microglia displayed an altered morphology, marked by greater process ramification (*Figure 5B* and *Figure 5—figure supplement 1*). Over the same 7-day time course, a greater proportion of microglial somata and processes displayed calcium activity, with increased calcium signaling lasting up to 10 days (*Figure 5C–E*). During this period, microglial somata and processes could sustain high amplitude (>2 fold $\Delta$F/F) calcium transients for exceptionally long periods (1–4 min; see $\Delta$F/F traces in *Figure 5C*). Calcium activity was also highly synchronized between the processes and soma, best exemplified by the appearance of large, spreading microglial calcium waves (*Figure 5D* and *Figure 5—Video 1*). For these reasons, it is not surprising that the proportion of active microdomains and their signal areas are highly similar during this 14-day period (*Figure 5E*). Both somata and processes showed large increases in calcium activity before returning to near-baseline levels two weeks later (*Figure 5E*). Only after the excitotoxic insult represented by kainate status epilepticus do we observe consistent increases in microglial somatic calcium activity and the appearance of coordinated, whole-cell calcium transients. The appearance of both phenomena suggest microglial calcium activity can be fundamentally changed following excitotoxicity.

In the days following kainate status epilepticus, microglial processes became exceptionally motile (*Figure 5F* and *Figure 5—figure supplement 2*), allowing us to study a number of motility events and their relationship with process calcium activity. One day after kainate status epilepticus, we find that 93% of microglial processes undergoing extension exhibit calcium activity, compared to 45% of stable processes and 38% of retracting processes (*Figure 5G*). Extending processes also exhibit calcium signaling nearly three-times the magnitude of stable or retracting processes (*Figure 5G* and *Figure 5—Video 2*). When charting the temporal relationship between a calcium event and the beginning of extension or retraction (*Figure 5F* and *Figure 5—figure supplement 2*), we find that 51% of extending processes have a closely aligned calcium event (within 60 s), compared to 18% of retracting processes (*Figure 5H*). Our analysis suggests that after status epilepticus, microglial process extension is temporally correlated and strongly associated with their calcium activity.

## DREADD-based modulation of excitatory neuronal activity is sufficient to induce microglial calcium signaling

Isoflurane- and kainate-based changes in neuronal activity represent two of the more extreme shifts in network function. We additionally employed chemogenetic approaches to achieve smaller, more direct alterations to network function. DREADDs (Designer Receptors Exclusively Activated by Designer Drugs) utilize the expression of exogenous G-protein receptors to modulate cellular activity (*Roth, 2016*). We used a Gi system (hM4D) targeted to local CaMKIIa neurons of the somatosensory cortex to decrease neuronal activity and a Gq system (hM3D) to increase neuronal activity (*Figure 6A*). In a first series of experiments, we tested multiple doses of the DREADD ligand, Clozapine N-oxide (CNO), with the goal of finding a dose that could produce modest (30–40%) and intermediate (60–70%) changes in excitatory neuronal calcium activity. We also examined the time of peak effect in both systems between 15 and 60 min after injection (*Figure 6A*, timeline). In the Gi system, a 2.5 mg/kg (i.p.) dose of CNO decreased CaMKIIa calcium signaling by 51 $\pm$ 4.2%, while a 5.0 mg/kg dose decreased signaling by 66 $\pm$ 3.5%. Due to the similar responses, a 1.25 mg/kg (i.p.)

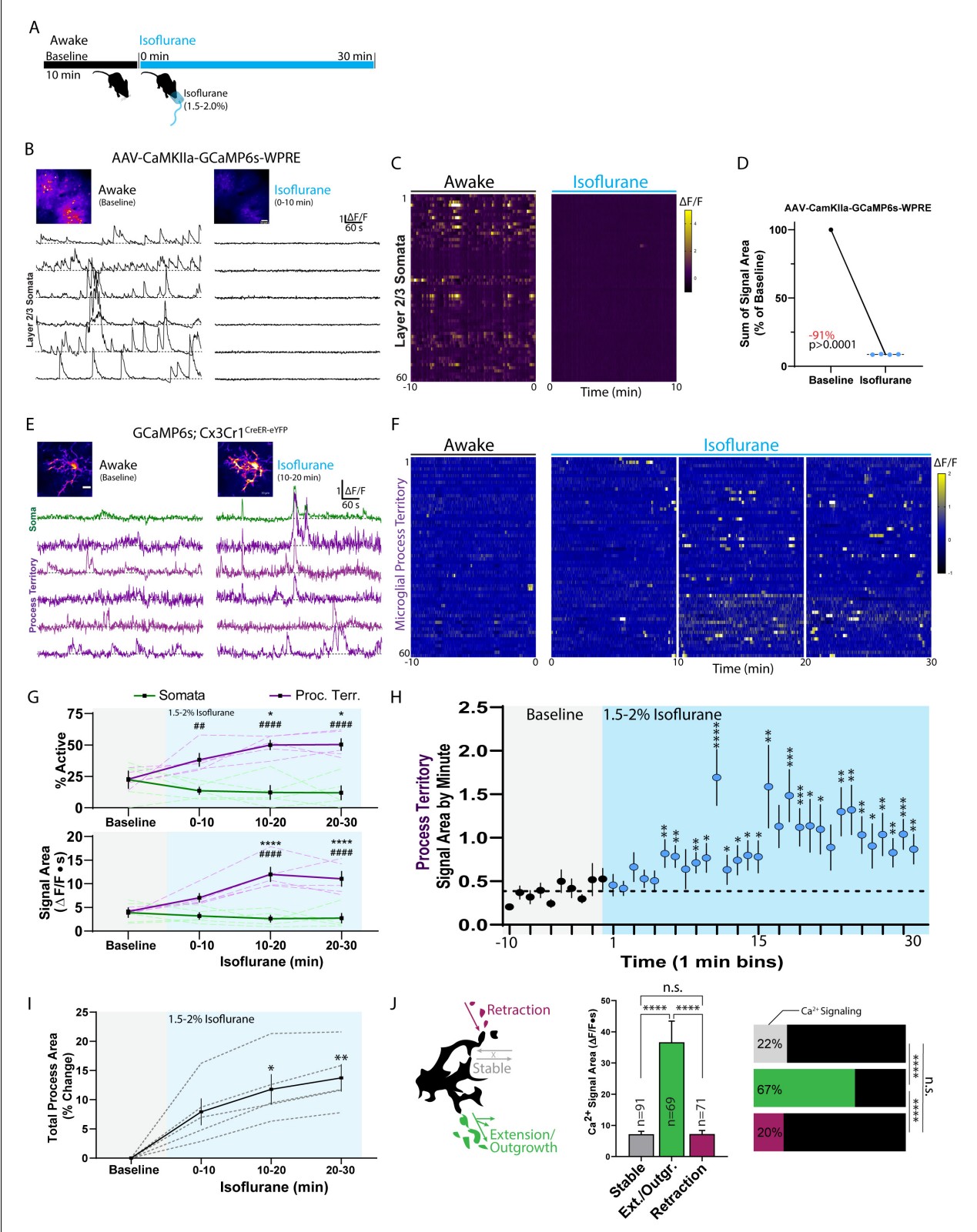

**Figure 3.** Isoflurane anesthesia increases microglial calcium activity. (A) Timeline of experiment in a chronic window animal. (B–D) Control studies of excitatory neuronal calcium activity before and after isoflurane induction. (B) Representative ΔF/F traces of somatic calcium activity from layer II/III CaMKIIa neurons in one animal. (C) A heatmap of CaMKIIa somatic calcium activity from 60 representative somatic ROIs in one animal under awake and isoflurane conditions. (D) Baseline-normalized neuronal calcium activity under isoflurane (Paired t-test; dots: one animal) showing an average 91% drop

*Figure 3 continued on next page*

*Figure 3 continued*

in calcium signaling in the first 10 min of isoflurane maintenance (1.5–2%). (E–H) Microglial calcium responses to isoflurane from a cohort of chronic window GCaMP6s;Cx3Cr1[CreER-eYFP] mice (see also: *Figure 3—Video 1*). (E) Representative ΔF/F traces under awake and isoflurane conditions from the depicted microglial cell using threshold-based segmentation. (F) A heat map of microglial calcium activity from 60 threshold-segmented process territories surveyed across multiple microglia in a single field of view from a representative animal. (G) Microdomains active and their signal areas from baseline through anesthesia (two-way ANOVA; *Dunnett's post-hoc vs. baseline; #somata vs. processes) using the threshold-segmentation approach for quantification. Dashed lines represent the averaged data from each of five animals, while solid lines represent the mean ± SEM from the N = 5 animals. (H) Microglial process territory calcium activity was binned by minute across all territories studied (311 to 379 territories detected in different imaging periods). The mean minute-by-minute signal area during the baseline period (0.385 ΔF/F•s, dashed line) was compared to minute-by-minute calcium signal areas during isoflurane exposure (one-sample t-test comparison to the baseline average), indicating significant calcium changes in microglial processes are detectable 6 min after exposure . (I) After masking the soma, uniform thresholding of the average intensity image by period shows increasing total process area during isoflurane exposure (dashed lines: individual animal; solid lines: group mean ± SEM; one-way ANOVA with Dunnett's post-hoc vs. baseline). (J) Microglial process calcium activity under isoflurane was studied by motility/morphology characteristics. As displayed in the cartoon, microglial motility/morphology was categorized as processes that retracted under isoflurane, processes that remained stable, or processes that extended, showing new areas of outgrowth. Exact criteria and methodology are provided in the methods and displayed further in *Figure 3—figure supplement 1*. After subdividing processes based on motility characteristics, the percentage of processes exhibiting calcium activity is displayed (horizontal bars, Fisher's exact test) along with the signal magnitude (one-way ANOVA with Tukey's post-hoc test; the number of sampled regions fitting each criterion is provided near the bar; mean ± SEM). Scale bars: 50 µm (B), 10 µm (E). N = 4 AAV-CaMKIIa-GCaMP6s mice (A–C), N = 5 GCaMP6s;Cx3Cr1[CreER-eYFP] mice (D–G). *p<0.05, **, ##p<0.01, ***p<0.001, ****, ####p<0.0001.

The online version of this article includes the following video, source data, and figure supplement(s) for figure 3:

**Source data 1.** Microglial calcium activity during isoflurane.

**Figure supplement 1.** Isoflurane anesthesia results in process extension/outgrowth with high calcium activity.

**Figure 3—video 1.** Microglial calcium changes in the awake mouse and during 30 min isoflurane administration.

https://elifesciences.org/articles/56502#fig3video1

dose of CNO was also tested, resulting in a 31 ± 2.7% decrease in CaMKIIa calcium activity. In the Gi system, peak reductions in excitatory signaling were reached 30 min after injection and were sustained for at least 30 min thereafter (*Figure 6B,D and E*). In the Gq system, a 2.5 mg/kg (i.p.) dose of CNO increased excitatory neuronal calcium signaling by 42 ± 6.5%, while a 5.0 mg/kg dose increased signaling by 70 ± 8%. In the Gq system, there was a clear peak effect 30–45 min after injection (*Figure 6C–E*). In a control study, the highest dose of CNO used (5 mg/kg) did not alter neuronal calcium signaling in mice without DREADD transfection (*Figure 6E*). Based upon these tests, we adopted a 1.25 and 5.0 mg/kg dose of CNO in the Gi system, and a 2.5 and 5.0 mg/kg dose in the Gq system to impart modest or intermediate effects, respectively.

To this end, GCaMP6s;Cx3Cr1[CreER-eYFP] mice were transfected with DREADD virus and microglial calcium activity was studied in the area of peak neuronal expression (mCherry tag; *Figure 6F*). In the Gi system, CNO-dependent decreases in CaMKIIa neuronal activity resulted in gradual increases in microglial process calcium activity, peaking 45–60 min after injection (*Figure 6G,I , J*, and *Figure 6—Video 1*). Notably, there were clear dose-dependent differences in the extent of microglial process calcium activity after CNO injection (*Figure 6I and J*), with a 64 ± 11% increase observed at the lower dose (1.25 mg/kg) and a 133 ± 26% increase observed at the higher dose (5.0 mg/kg). In the Gq system, microglial process calcium activity displayed highly similar trends to the Gi system, despite the fact that these approaches resulted in opposing shifts to neuronal activity (hypoactivity in the Gi system and hyperactivity in the Gq system). At the lower dose in the Gq system, 2.5 mg/kg CNO administration resulted in a 68 ± 31% increase in microglial process calcium activity, while the higher dose (5.0 mg/kg) resulted in a 128 ± 13% increase in process calcium activity (*Figure 6H–J*, and *Figure 6—Video 1*). In both the Gi and Gq systems, peak microglial calcium responses occurred 45–60 min after CNO administration, suggesting an approximate 15 min latency relative to peak neuronal changes (*Figure 6D and I*). Additionally, both Gq- and Gi-based alterations to excitatory neuronal activity did not result in observable changes to microglial somatic calcium signaling (data not shown). Similar to isoflurane and kainate experiments, microglial process extension/outgrowth was the predominant motility behavior observed following Gi- or Gq-DREADD activation in neurons (*Figure 6—figure supplement 1*). Extension was the motility behavior most closely associated with calcium activity and large signal magnitude (*Figure 6K,L*, and *Figure 6—figure supplement 1*). Overall, we observe that a near-linear range of neuronal activity shifts (*Figure 6E*) results in a

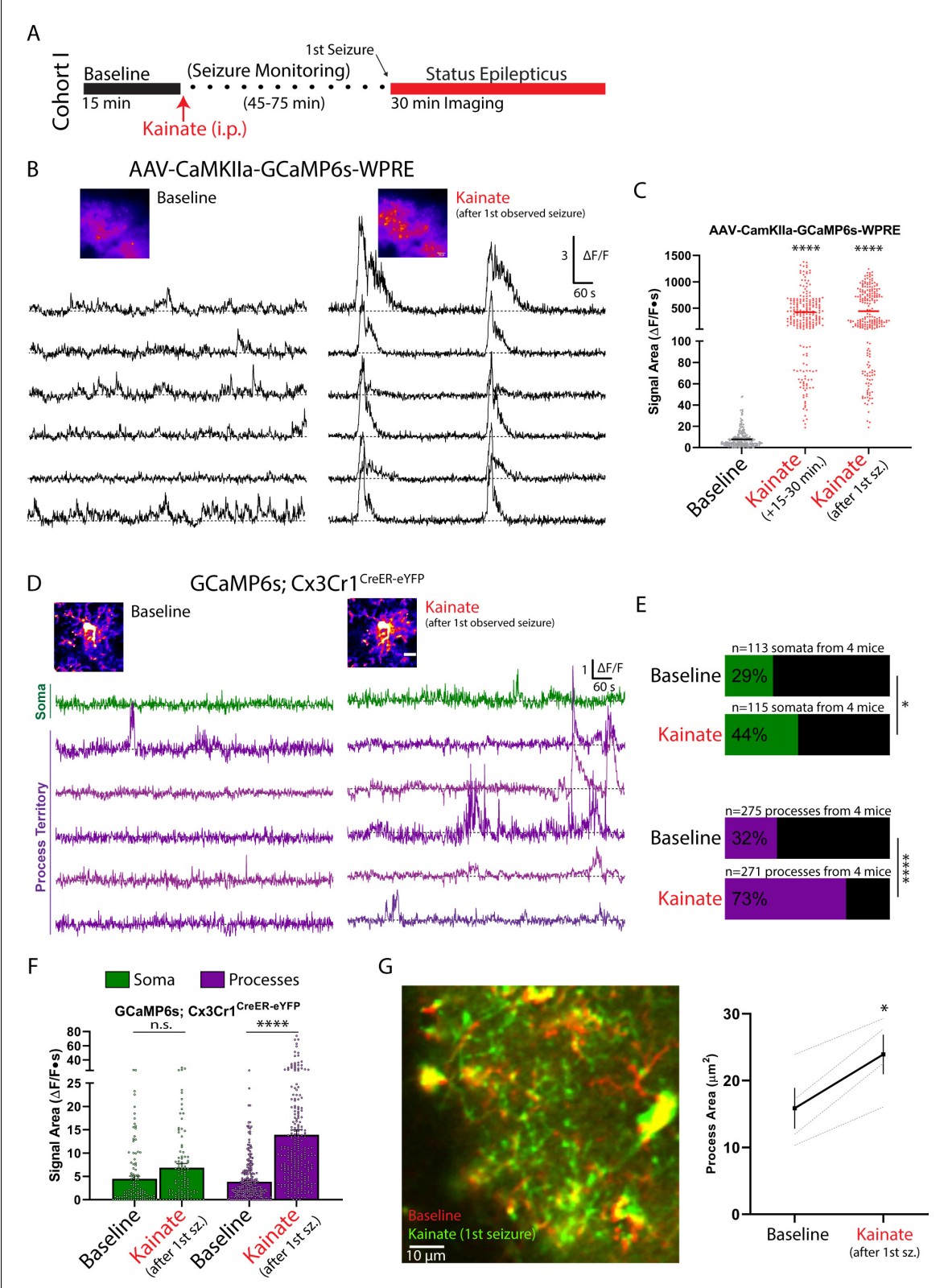

**Figure 4.** Increased microglial process calcium activity immediately following the observation of kainate-induced seizures. (A) Outline of the experimental timeline. Mice experienced a first seizure 45–75 min after kainate injection, which marked the beginning of imaging during periods labelled as 'kainate.' (B–C) CaMKIIa excitatory neuronal calcium activity (AAV.CaMKIIa.GCaMP6s.WPRE transfection) in the somatosensory cortex before and after kainate administration. (B) Representative ΔF/F traces at baseline, and after the first observed generalized seizure (45–75 min after kainate). (C)

*Figure 4 continued on next page*

Figure 4 continued

Signal areas derived from CaMKIIa somata in layer II/III (1-Way ANOVA with Dunnett's post-hoc comparison to baseline). (D-G) Microglial calcium activity (GCaMP6s;Cx3Cr1CreER-eYFP) in the awake animal at baseline and after the first kainate-induced generalized seizure. (D) Representative ΔF/F traces from a microglial cell using threshold-based ROI detection. (E) The percentage of active microdomains under each condition (Fisher's exact test). (F) Summary of calcium signal area values at baseline and after the first generalized seizure (two-way ANOVA with post-hoc comparison between kainate and baseline), using the threshold-segmentation approach for quantification. (G) Two-color, average intensity images of microglial morphology at baseline and after kainate-induced seizures, suggesting overall process extension. Changes in process area are quantified between conditions (paired t-test). Scale bars: 50 µm (B), 10 µm (D). Grouped data represent the mean ± SEM; dots represent individual neuronal somata (C) or individual microdomains (F); dashed lines represent individual animals (G). N = 3 mice for neuronal calcium studies (B, C), N = 4 mice for microglial calcium studies (D–F); the number of microdomains surveyed is provided in (D). *p<0.05, ****p<0.0001.

The online version of this article includes the following source data for figure 4:

**Source data 1.** Microglial calcium activity during kainate status epilepticus.

---

U-shaped curve of microglial process calcium activity (*Figure 6J*), with the greatest shifts in neuronal activity producing the largest microglial process calcium signaling (*Figure 7*).

## Discussion

Herein, we describe a series of observations regarding microglial calcium signaling in response to shifts in neuronal activity. In general, microglia are considered to have the lowest levels of spontaneous calcium activity in 'resting' or 'baseline' states from reports in anesthetized mice (*Brawek et al., 2017*; *Eichhoff et al., 2011*; *Pozner et al., 2015*). We observe that microglial calcium activity is similarly low in awake, chronic window mice (*Figures 1* and *2*). However, pharmacological alterations in neuronal activity can trigger microglial process calcium signaling without impacting somatic calcium signaling (*Figures 3*, *4* and *6*). In general, microglial process calcium signaling follows a U-shaped curve, with greater shifts in neuronal activity (both hypoactive or hyperactive) resulting in stronger process calcium signaling (*Figure 7*). Additionally, both shifts in network activity were associated with microglial process extension/outgrowth. Extending processes consistently demonstrated the greatest calcium activity. Our work initially suggests that microglial calcium signaling may be an important component in how microglia address brain state changes.

### Spontaneous microglial calcium activity and the utility of different GCaMP variants and preparations

Microglial have low spontaneous calcium activity in the anesthetized animal (*Brawek et al., 2017*; *Eichhoff et al., 2011*; *Pozner et al., 2015*), relative to reports of calcium activity in neurons and astrocytes. We initially wanted to determine if low spontaneous calcium activity in earlier reports was a product of the anesthesia used during imaging. Even in the awake animal, microglia do indeed display very infrequent spontaneous process and soma calcium activity (*Figures 1* and *2*). Previous studies also arrive on a key concept that injury, damage, and inflammation all produce increased or altered microglial calcium signaling (*Brawek et al., 2017*; *Brawek et al., 2014*; *Eichhoff et al., 2011*; *Pozner et al., 2015*). As resident CNS immune cells, microglia have a strong ability to sense inflammation and even low-level injury, which can occur during a craniotomy. By some reports, microglia can maintain a more reactive phenotype for multiple weeks after craniotomy (*Xu et al., 2007*). For this reason, it was important that we compare microglial calcium signaling in an acute and chronic window preparation (*Figure 2*). We found multiple lines of evidence that microglial process calcium signaling, but not somatic signaling, may be artificially inflated in the acute window preparation. In acute window preparations, a higher proportion of processes exhibited calcium signaling, and a subset of processes displayed calcium signals that were orders of magnitude above the average. In chronic window animals, both of these phenomena were attenuated. However, we could still evoke larger or widespread process calcium activity in chronic window animals, but only after introducing local or systemic changes in circuit activity (i.e. kainate). These observations suggest that chronic window preparations may better reflect the occurrence and magnitude of true spontaneous microglial calcium activity. The stability of spontaneous calcium activity in this preparation over multiple days also suggests that it is ideal for longitudinal studies.

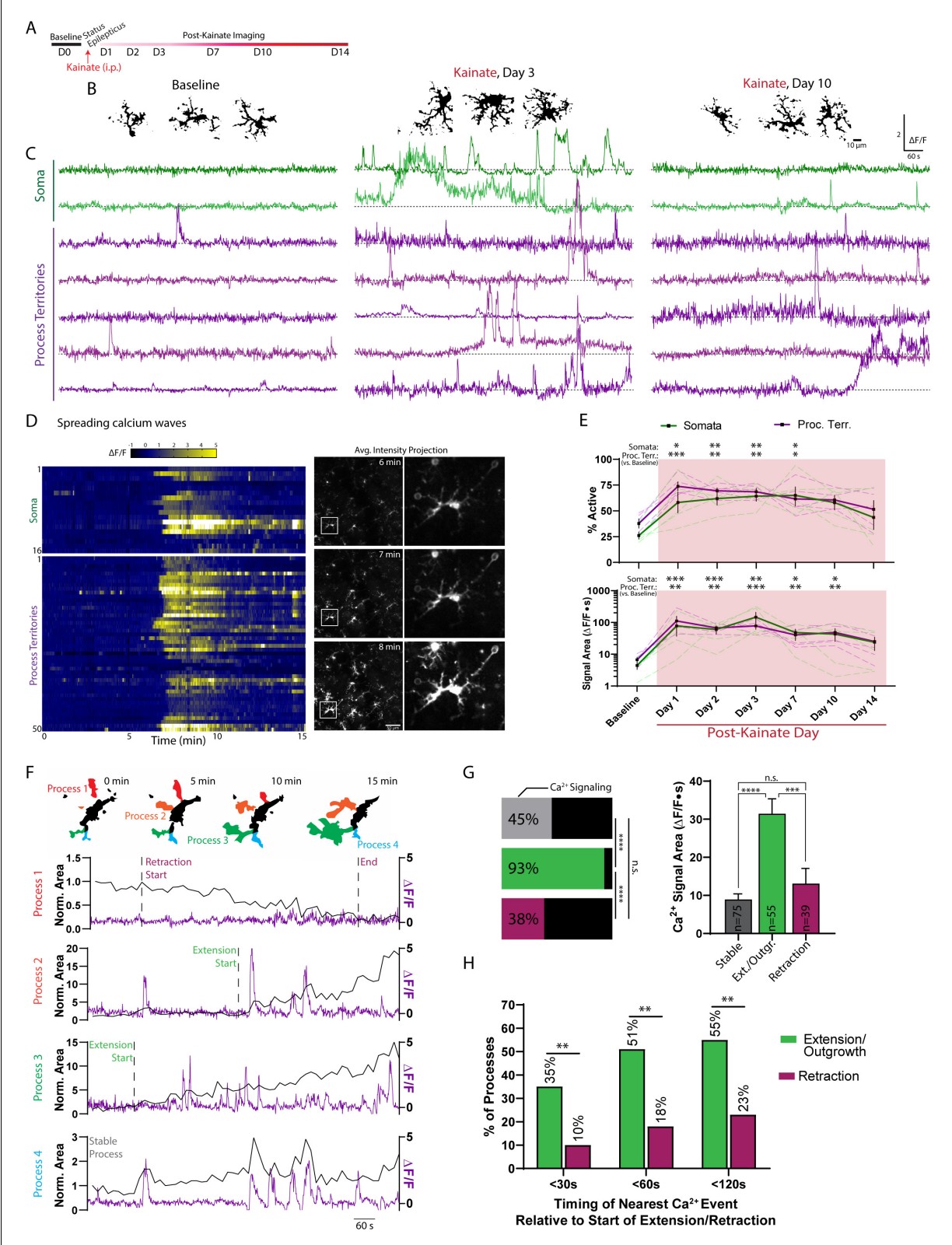

**Figure 5.** Kainate administration leads to longitudinal modulation of microglial calcium signaling. (A) Timeline of experiment. (B) Representative images of microglia morphology at baseline and following kainate status epilepticus. See also: *Figure 5—figure supplement 1*. (C) ΔF/F traces of microglial somatic and process calcium activity, using the threshold-based approach to detect microglial process territories. (D) Heat map of a microglial spreading calcium wave one day after kainate status epilepticus. Each row represents activity from a soma or one of 50 threshold-segmented

*Figure 5 continued on next page*

*Figure 5 continued*

process territories surveyed across this single field of view and animal. See also: *Figure 5—Video 1*. Corresponding images of microglia across the whole field of view during the peak of wave activity, with a highlighted cell magnified. (E) Microdomains active and their signal areas (two-way ANOVA with Dunnett's post-hoc vs. baseline). (F) Representative changes in microglial process area (black line; 15 s avg. intensity images) correlated with ΔF/F calcium activity (purple line). Changes in process area were used to denote the start and end of retraction or extension. See also: *Figure 5—figure supplement 2*. (G) Based upon these criteria, extending processes were more likely to have associated calcium activity than processes undergoing retraction or remaining stable (horizontal bars, Fisher's exact test). Additionally, the calcium signals observed in extending processes were larger in magnitude (one-way ANOVA with Tukey's post-hoc test; the number of sampled regions fitting each criterion is provided in the bar; mean ± SEM). See also: *Figure 5—Video 2*. (H) The percentage of processes exhibiting a calcium transient within a certain time window relative to the start of process extension or retraction. This analysis considers the closest temporally aligned calcium event preceding and/or following the start of extension/retraction (see methods for further details; survey of 55 extending processes and 39 retracting processes; Fisher's exact test). Scale bar: 50 μm (D). N = 5 GCaMP6s;Cx3Cr1$^{CreER-eYFP}$ mice. Grouped data represent mean ± SEM; dashed lines represent individual animals (E). *p<0.05, **p<0.01, ***p<0.001. The online version of this article includes the following video, source data, and figure supplement(s) for figure 5:

**Source data 1.** Longitudinal changes in microglial calcium activity following kainate status epilepticus.
**Figure supplement 1.** Full Sholl analysis quantification of morphology changes, related to panel 5B.
**Figure supplement 2.** One day after kainate status epilepticus, process extension is closely coordinated with microglial calcium activity.
**Figure 5—video 1.** Spreading microglial calcium waves recorded 1 day after kainate status epilepticus.
https://elifesciences.org/articles/56502#fig5video1
**Figure 5—video 2.** Microglial process extension co-occurring with increased calcium activity.
https://elifesciences.org/articles/56502#fig5video2

One of the more unique observations made in our initial studies is the low-level, sometimes widespread fluorescent signature reported by the Lck-GCaMP6f animal (*Figure 1E*). We initially investigated calcium signaling using the Lck-GCaMP6f animal, because tethering GCaMP to the cell membrane could reveal key calcium signals in fine process structures, as demonstrated for astrocytes (*Srinivasan et al., 2015*). We hypothesize that the coordinated, low-level signals detected by the Lck-GCaMP6f mouse could reflect neuronal activity changes sensed by surrounding microglial processes. In future studies, we will use bipolar EEG recording inside of the window to determine if the widespread, low-level fluorescent signature reported by Lck-GCaMP6f indicator is temporally well-aligned with increased cortical activity. If so, this approach could provide the first evidence that microglia can respond to neuronal activity in real time. Prior reports have demonstrated real-time calcium responses only in the presence of pathology or injury (*Eichhoff et al., 2011*).

An interesting question is whether the different GCaMP variants may also be able to report different mechanisms for calcium elevations. Given its cytosolic localization, GCaMP6s may be better suited to detect calcium release from the endoplasmic reticulum, which is canonically attributed to Gq-coupled GPCR signaling. On the other hand, the membrane localization of Lck-GCaMP6f may be able to detect calcium entry through membrane channels. Microglia express both ionotropic and metabotropic receptors for calcium elevations, particularly in the P2X (e.g. P2X7, *Eyo et al., 2013*) and P2Y (e.g. P2Y6, *Koizumi et al., 2007*) family. Pharmacology experiments in vivo suggest that both routes of calcium entry are potentially viable in microglia (*Eichhoff et al., 2011*), but cytosolic calcium levels may be more important in damage responses (*Eichhoff et al., 2011*; *Pozner et al., 2015*). On the other hand, our previous work also suggests that extracellular calcium inhibits microglial process convergence, a key response to neuronal hyperactivity (*Eyo et al., 2015*). Future studies will investigate the potential signaling mechanism(s) underlying microglial calcium elevations in response to neuronal hyperactivity and hypoactivity.

## The relationship between neuronal activity changes, process extension, and process calcium signaling

Both approaches to increase or decrease neuronal activity resulted in elevated microglial calcium activity after a 6 to 15 min latency. The nature of the latency between neuronal changes and microglial calcium signaling does not suggest that microglia have a real-time response to neuronal activity in these instances. Instead, microglia appear to mobilize calcium signaling some minutes later. Our results suggest that microglial calcium signaling after neuronal activity changes is correlated with process motility, and extension in particular. This could reasonably explain the observed latency between neuronal activity and microglial calcium signaling. Our previous report suggests that

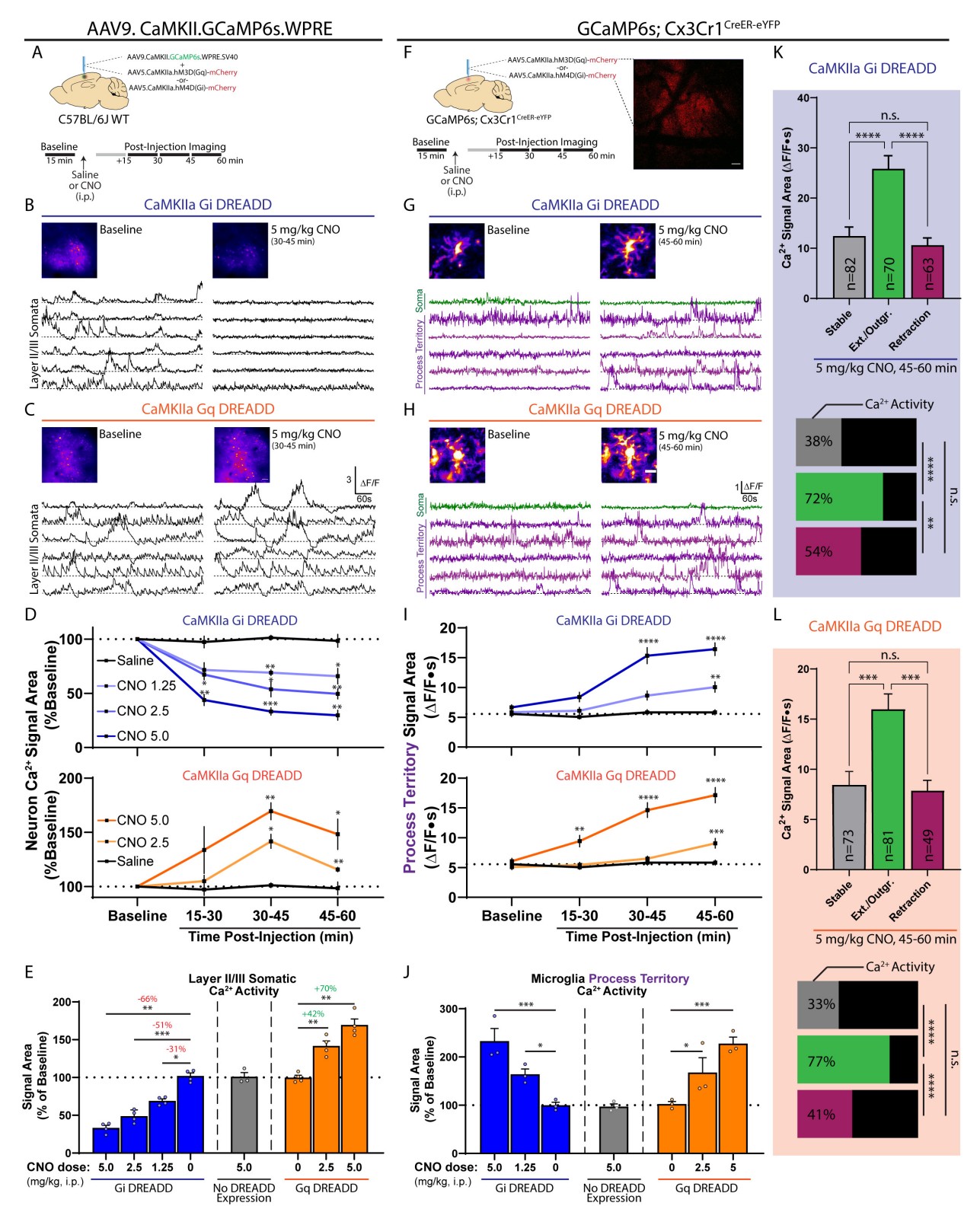

**Figure 6.** DREADD-based modulation of excitatory neuronal activity is sufficient to induce microglial calcium signaling. (**A**) AAVs injected into the somatosensory cortex of WT mice in order to study DREADD-based changes in neuronal calcium activity (top). Experiment outline (bottom). (**B–C**) Representative ΔF/F traces of layer 2/3 CaMKIIa neuronal calcium activity at baseline and after CNO injection in the Gi (**B**) and Gq (**C**) systems. (**D**) Saline or CNO dose-dependent effects on neuronal calcium activity in the Gi and Gq systems over 15 min imaging periods (one-way ANOVA with

*Figure 6 continued on next page*

*Figure 6 continued*

Dunnett's post-hoc comparison to saline). (E) Summary of DREADD-based effects on neuronal calcium activity (at the time of peak effect: 30–45 min post injection; two-way ANOVA with Dunnett's post-hoc comparison to baseline). (F) Method to study DREADD-based changes in neuronal activity on microglial calcium signaling. mCherry expression served as a positive control for successful DREADD transfection. (G–H) Representative ΔF/F traces of microglial soma and process territory calcium activity at baseline and after CNO injection in the Gi (G) and Gq (H) systems. The threshold-based ROI approach was used to detect process regions. (I) Microglial process territory calcium activity in the Gi and Gq systems over 15 min imaging periods (one-way ANOVA with Dunnett's post-hoc vs. saline). (J) Summary of DREADD-based effects on microglial process territory calcium activity (at the time of peak effect: 45–60 min post injection; two-way ANOVA with Dunnett's post-hoc comparison to baseline). (K) The mean ± SEM calcium signal area is plotted for microglial process territories that remained stable, demonstrated new process outgrowth, or exhibited retraction 45–60 min after 5 mg/kg CNO injection in the Gi DREADD system (one-way ANOVA with Tukey's post-hoc test; process sample size given in the bar). The percentage of processes exhibiting calcium activity by motility/structural characteristic is displayed in the horizontal bars (Fisher's exact test). (L) The same analyses as performed in (K) for the Gq DREADD system (see *Figure 6—figure supplement 1* for more detail). Scale bars: 50 µm (B, C, and F), 10 µm (G-H). (A-E) N = 8 WT mice, N = 4/8 receiving AAV-CaMKIIa-hM4D(Gi) and N = 4/8 receiving AAV-CaMKIIa-hM3D(Gq). (F-J) N = 6 GCaMP6s;Cx3Cr1[CreER-eYFP] mice, half receiving Gq or Gi DREADD. Grouped data represent mean ± SEM; dots represent individual animals (E, J). *p<0.05, **p<0.01, ***p<0.001, ****p<0.0001. See also: *Figure 6—Video 1*.

The online version of this article includes the following video, source data, and figure supplement(s) for figure 6:

**Source data 1.** Microglial calcium activity during Gi and Gq DREADD activation in CaMKIIa neurons.
**Figure supplement 1.** CaMKIIa Gi and Gq activation results in microglial process extension and coordinated process calcium activity.
**Figure 6—video 1.** Microglial calcium activity at baseline and following CNO administration in Gi and Gq systems.
https://elifesciences.org/articles/56502#fig6video1

isoflurane anesthesia applied to the awake mouse engages maximal process movement after a delay of up to 10 min (*Liu et al., 2019*), and our process area analysis in the current study replicates these findings (*Figure 3H*). On a similar timescale to process extension/outgrowth, microglia begin to show elevated process calcium signaling (6 min delay). We posit that the calcium associated with extending processes contributes to the majority of the overall calcium signaling observed. Similarly, we observe elevated microglial process calcium activity after Gq or Gi DREADD activation in CaMKIIa neurons. In these studies, peak microglial calcium activity was delayed by approximately 15 min from the timing of peak of neuronal activity. This longer delay may be attributable to the more gradual alterations in neuronal activity produced by DREADD-based approaches or the lesser overall magnitude of changes observed (30% to 70% change in neuronal calcium activity relative to an immediate 90% decrease under isoflurane). In either case, the timing of microglial process calcium activity follows the time course of motility changes, with extending processes contributing the most to overall calcium increases. This time scale is also on par with other microglial process motility responses, such as laser burn or local ATP, which require 10–20 min for processes to fully extend (*Davalos et al., 2005*; *Wu et al., 2007*).

The relationship between motility and calcium signaling in microglia, first observed in culture (*Langfelder et al., 2015*; *Nolte et al., 1996*), is complex. Extending processes are not categorically associated with calcium activity in the present study (67–93% of extending processes had calcium activity across experiments). The temporal relationship between extension and calcium activity is also not universal, with approximately 50% of extending processes having a closely timed calcium event when studied after kainate status epilepticus (*Figure 5H*). As a ubiquitous secondary messenger, calcium signaling is not expected to perfectly align with any one cellular function, such as motility. Our results suggest that calcium may be an important contributor to extension. Previous studies in vivo have shown that chelating calcium in microglia can reduce process velocity, but not abolish movement entirely (*Pozner et al., 2015*). Thus, calcium could be a co-regulator of motility under certain conditions, potentially modulating the P2Y12/Gi-based cAMP pathway or PI3 kinase signaling canonically implicated in process movement (*Haynes et al., 2006*; *Wu et al., 2007*).

Another key future consideration is whether microglial calcium signaling occurs specifically when neuronal activity shifts away from baseline specifically, or shifts in any manner (such as a shift from hypoactive back to baseline). We would hypothesize that shifts back to baseline may not be accompanied by microglial calcium activity, based upon the types of motility associated with these shifts. Specifically, we posit that process retraction would be most prominently observed during shifts back to baseline, and retracting processes rarely appear to mobilize calcium activity. These experiments

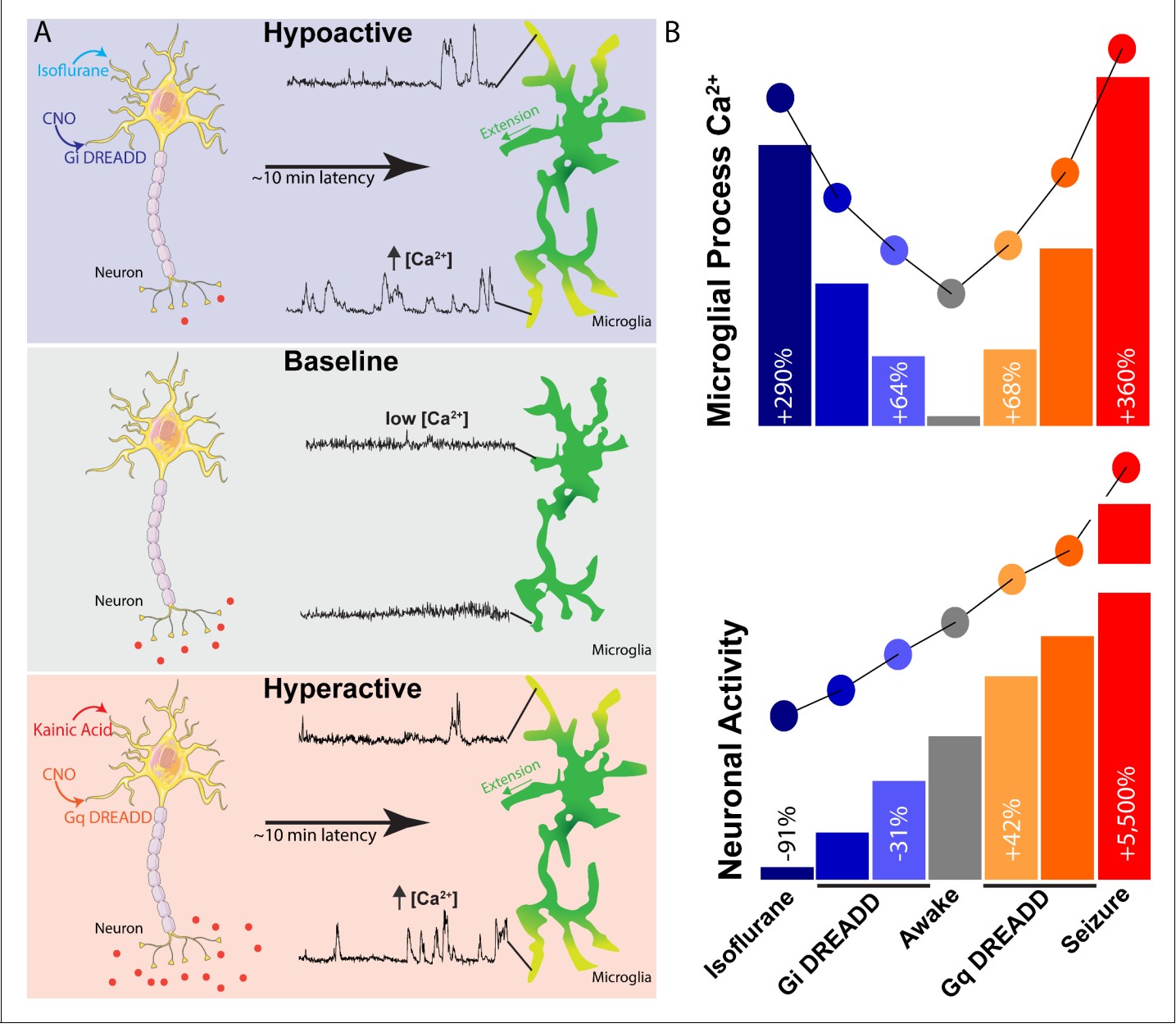

**Figure 7.** Summary of findings and observed relationship between neuronal activity and microglial calcium signaling. (**A**) During 'baseline' levels of neuronal activity in the awake mouse (middle panel), microglia demonstrate the lowest level of process calcium activity. However, shifts towards neuronal hypoactivity (isoflurane or Gi DREADD activation, top panel) or hyperactivity (kainate or Gq DREADD activation, bottom panel), result in microglial process extension and increased process calcium signaling after a multi-minute latency. (**B**) A near-linear range of neuronal activity (recorded as cortical layer II/III somatic calcium activity; bottom) results in a U-shaped curve of microglial process calcium increases (top panel). These data represent the average signal areas recorded across experiments.

will be ideal for testing predictions made from our early observations about the relationship between different motility behaviors and calcium signaling.

### The importance of microdomains in understanding glial calcium activity

Our findings clarify the role of neuronal activity on microglial calcium signaling. Previous studies using the GABA antagonist bicuculline to increase neuronal firing have come to different

conclusions. In a first study, bicuculline did not alter microglial calcium activity (*Eichhoff et al., 2011*). However, this earlier study investigated bicuculline effects largely by imaging microglial somatic calcium activity reported through a calcium indicator dye. Our work suggests that it is unlikely that acute changes in neuronal activity, even under seizure conditions, will dramatically impact microglial somatic calcium activity. In a more recent study using the calcium indicator GCaMP5G, microglia exhibited strong calcium wave activity when bicuculline was applied (*Pozner et al., 2015*). Notably though, changes in neuronal activity were evoked in an inflammatory context, specifically occurring 12–24 hr after LPS application, which suggests a synergy between inflammation and excitation on microglial calcium activity. We hypothesize that this observation is informative for interpreting our longitudinal studies following kainate status epilepticus. The period following a brain insult like status epilepticus alters neuronal network synchronization, but it can also provoke inflammation (*Bar-Klein et al., 2014*; *Kim et al., 2017*; *Tian et al., 2017*; *van Vliet et al., 2018*). Under these inflammatory and epileptogenic conditions, we similarly observe wave-like calcium activity in approximately 10% of imaging sessions, similar to the frequency described with bicuculline and LPS (*Pozner et al., 2015*). Notably, only under these long-term, pro-inflammatory conditions do we observe clear changes in microglial somatic calcium activity. The profoundly altered calcium signaling properties of microglia following status epilepticus, including whole-cell calcium transients, suggests that calcium may serve as a key secondary messenger system in microglia during disease development.

An important consideration in our work is the potential impact of the low-level eYFP signal. This signal was consistently detected alongside dynamic changes in GCaMP6 fluorescence, contributing to a higher baseline ($F_0$) in $\Delta F/F_0$ calculations. For these reasons, our work could underestimate the amount of calcium activity in microglia. Additionally, the eYFP signal could mask small, but biologically important calcium signals, like those reported in astrocytes (*Srinivasan et al., 2015*). This could be most prominent in the soma, which exhibits a stronger eYFP signal. However, our results do not suggest such a caveat is likely to influence our major findings, as detected soma events were large in amplitude, prolonged, and exceedingly rare in the absence of pathology. In addition, these conclusions are very similar to other in vivo studies (*Eichhoff et al., 2011*; *Pozner et al., 2015*), which recorded somatic microglial calcium activity in the absence of an eYFP signal. Of interest, our microdomain analysis suggest a similar concept in how both astrocytes and microglia utilize calcium activity. Astrocytes are known to exhibit calcium signaling most often in their fine processes, with somatic elevations being relatively less common (*Taheri et al., 2017*). Thus, in both cases, it appears glial cells exhibit more calcium activity in their smaller process domains—the regions most likely to interact with synapses, other glia, or vasculature (*Zhao et al., 2018*).

The mechanisms regulating microglial calcium signaling have not been clearly identified. ATP/ADP is the most definitive agonist of microglial calcium signaling to date, and purinergic receptors (ionotropic P2X and metabotropic P2Y family) have been clearly implicated in transducing microglial calcium signaling (*Eichhoff et al., 2011*; *Färber and Kettenmann, 2006*; *Pozner et al., 2015*). While kainate and the Gq DREADD system could reasonably increase the release of purinergic signaling molecules (*Engel et al., 2016*), evoking both process outgrowth (canonically P2Y12-driven, *Eyo et al., 2014*; *Haynes et al., 2006*; *Swiatkowski et al., 2016*) and process calcium activity (potentially P2Y6-driven, *Koizumi et al., 2007*), it would be illogical for Gi DREADD and isoflurane to increase purinergic signaling. Recent studies suggest that depressing neuronal activity in the awake mouse prevents noradrenergic signaling to microglial β2 receptors (*Liu et al., 2019*; *Stowell et al., 2019*). During awake imaging, β2 receptor activation in microglia restrains process surveillance, motility, and extension, keeping microglia in a relatively arrested state. The absence of this signaling axis when neuronal activity is reduced allows for extension, as we observed here in isoflurane and Gi DREADD studies. What is unclear, however, is whether β2 receptor disinhibition also contributes to microglial calcium signaling during neuronal hypoactivity. In future, it will be critical to determine the mechanisms that govern microglial calcium responses to bidirectional shifts in neuronal activity. It is indeed possible that microglia indirectly respond to neuronal activity shifts, potentially through other cell types, like astrocytes, or even associated vascular changes; however, DREADD-based experiments do suggest that direct neuronal activity shifts are upstream of microglial calcium elevations. Overall, we demonstrate that microglia display increases in process calcium signaling following bidirectional shifts in neuronal activity. Our work leaves open the possibility that

microglial calcium signaling represents an important component in how microglia address brain state changes.

# Materials and methods

**Key resources table**

| Reagent type (species) or resource | Designation | Source or reference | Identifiers | Additional information |
|---|---|---|---|---|
| Strain, strain background (*M. musculus*; male and female) | C57BL/6J | Jackson labs | #016962 | |
| Strain, strain background (*M. musculus*; male and female) | CX3CR1[CreER-IRES-eYFP] | Jackson labs | #021160 | |
| Strain, strain background (*M. musculus*; male and female) | GCaMP6s (cytosolic) | Jackson labs | #024106 | |
| Strain, strain background (*M. musculus*; male and female) | Lck-GCaMP6f | Jackson labs | #029626 | |
| Transfected construct (*M. musculus*) | pENN.AAV.CaMKII.GCaMP6f.WPRE.SV40 | Addgene | Cat #: 105537-AAV9; RRID:Addgene_100834 | |
| Transfected construct (*M. musculus*) | pAAV.CaMKIIa.hM3D(Gq)-mCherry | Addgene | Cat #: 50476-AAV5; RRID:Addgene_50476 | |
| Transfected construct (*M. musculus*) | pAAV.CaMKIIa.hM4D(Gi)-mCherry | Addgene | Cat #: 50477-AAV5; RRID:Addgene_50477 | |
| Chemical compound, drug | Clozapine N-Oxide | Cayman | Cat #: 16882 | |
| Chemical compound, drug | Isoflurane | Terrell | Cat #: 66794-019-10 | |
| Chemical compound, drug | Kainic acid | Tocris | Cat #: 0222 | |
| Chemical compound, drug | Tamoxifen chow | Envigo | Cat #: TD.130856 | |
| Software, algorithm | Excel | Microsoft | | |
| Software, algorithm | ImageJ | NIH | | Java 1.8.0_172 |
| Software, algorithm | Prism | GraphPad | | Version 8 |
| Other | 5.2 mm Headplate | Neurotar | Model 2 | |
| Other | Closed-loop temperature controller | Physitemp | TCAT-2DF | Used during isoflurane imaging |
| Other | VivoScope Galvo Multiphoton | Scientifica | | |

*Continued on next page*

*Continued*

| Reagent type (species) or resource | Designation | Source or reference | Identifiers | Additional information |
|---|---|---|---|---|
| Other | Mai-Tai DeepSee | Spectra-Physics | MAI TAI HP DS | 2-P laser |
| Other | Mobile HomeCage | Neurotar | Cat #: NTR000289-01 | Air table for in vivo imaging |
| Other | UMP3 Ultra Micro Pump | WPI | Cat #: UMP3-4 | Used in AAV delivery |

## Mice

GCaMP6s (Rosa26-CAG-LSL-GCaMP6s; 024106), Lck-GCaMP6f (Rosa26-CAG-LSL-Lck-GCaMP6f; 029626), and CX3CR1$^{CreER-eYFP}$ (021160) knock-in mouse lines can be obtained from the Jackson Laboratory. Neuronal calcium activity was studied in C57BL/6J WT mice using AAV injections. Both male and female offspring were used across all studies at an age ranging from 3 to 5 months. Mice were group housed in an AAALAC-approved facility in climate-controlled rooms with a 12 hr light/dark cycle (lights on at 6am) and had free access to food and water. All experimental procedures were approved by the Mayo Clinic's Institutional Animal Care and Use Committee (IACUC).

## Administration of tamoxifen in chow

In CreER lines, tamoxifen was administered through chow to activate GCaMP expression in microglia. Mice were weaned at P21 and then provided tamoxifen in chow for a two-week period (250 mg tamoxifen per 1 kg of chow; Envigo). Notably, chronic window studies occurred no sooner than 4 weeks after tamoxifen administration ended, which is sufficient to label microglia but not peripheral CX3CR1 cell types due to differences in their turnover rates (*Parkhurst et al., 2013*).

## Stereotaxic delivery of AAV

Under isoflurane anesthesia (4% induction, 1.5–2.5% maintenance), AAVs were injected into the somatosensory cortex (AP: −4.5, ML: +2.0) using a glass pipette and micropump (World Precision Instruments). AAVs were targeted to both layer V neurons (DV: −0.5) and layer II/III neurons (DV: −0.2). A 250 nL volume was dispensed at each level at a 40 nL/min rate followed by a 10-min rest period for diffusion. pENN.AAV9.CaMKII.GCaMP6s.WPRE.SV40, AAV5.CaMKIIa.hM3D(Gq)-mCherry, and AAV5.CaMKIIa.hM4D(Gi)-mCherry were either injected alone (1:1 dilution in PBS) or in conjunction (1:1 ratio of DREADD and GCaMP virus). All viruses were used at a titer of $10^{12}$ GC/mL and acquired from Addgene.

## Cranial window surgery

Cranial windows were implanted following standard techniques. Under isoflurane anesthesia (4% induction, 1.5–2.5% maintenance), a circular craniotomy (<3 mm diameter) was made over somatosensory cortex (AP: −4.5, ML: +2.0) using a high-speed dental drill. A circular glass coverslip (3 mm diameter, Warner) was secured over the craniotomy using Loctite 401 at the lateral edges. A four-point headbar (NeuroTar) was secured over the window using dental cement. A mild NSAID solution (0.2 mg/mL Ibuprofen) was provided in the home cage for 72 hr following surgery. In mice undergoing AAV injections, windows were implanted over the injection site after a 3-week recovery period.

## Two-photon imaging: acute and chronic imaging parameters

Imaging in the awake animal was performed using a Scientifica multiphoton microscope equipped with galvanometer scanning mirrors. To image GCaMP fluorescence, a Mai-Tai DeepSee laser was tuned to 920 nm and kept below 50 mW output for layer I imaging or 55 mW output for layer II/III imaging, as measured directly under the objective. Both the eYFP and GCaMP6s signals were passed through a 520/15 filter (Chroma). Imaging occurred at a 1 Hz frame rate at 512 × 512 pixel resolution using a 16x water-immersion objective (NA: 0.8, Nikon). Microglial calcium was imaged at a consistent zoom factor (300 × 300 μm area), while neuronal calcium was sampled from a larger area (450 × 450 μm).

### Two-photon imaging: training and baseline imaging (*Figure 2*)

Mice were trained to move on an air-lifted platform (NeuroTar) while head-fixed under a 2P objective. Training occurred for 30 min/day for the first 3 days following surgery and once per week thereafter. On the first day following surgery, a subset of mice was imaged to determine baseline microglial calcium activity in the acute window. Chronic imaging occurred no sooner than 4 weeks after surgery in mice with a clear window. Across all studies, mice were allowed 10 min to acclimate after being placed in the head restraint before imaging began. For studies of spontaneous calcium activity in awake acute and chronic window animals, a 15-min video was acquired 55–75 µm below the dura (layer I) and 150–170 µm below the dura (layer II/III).

### Two-photon imaging: isoflurane studies (*Figure 3*)

A cohort of chronic window GCaMP6s;Cx3Cr1$^{CreER}$ mice and AAV9.CaMKII.GCaMP6s WT mice was first imaged under awake, baseline conditions (10 min; 55–75 µm depth for microglia, 150–170 µm depth for neurons). A nose cone was then secured to the head-restraint system and the mouse was induced with isoflurane on the platform (4%, up to 60 s). Isoflurane was maintained at 1.5–2% for the duration of imaging, acquired as three 10 min blocks. Under isoflurane, body temperature was maintained at 37°C (Physitemp).

### Two-photon imaging: kainate studies (*Figures 4* and *5*)

Two separate cohorts of GCaMP6s;Cx3Cr1$^{CreER}$ mice were used to study either the acute effects of kainate during status epilepticus (*Figure 4*) or the longitudinal effects of KA-SE over a 14 d period (*Figure 5*). To study the acute effects of kainate status epilepticus, a cohort of chronic window mice were imaged at baseline (15 min, 60–75 µm depth) then removed from the stage and injected with 19 mg/kg kainate (i.p.). Mice were then visually monitored for a generalized seizure (defined by Racine stage 3–5 criteria; *Racine, 1972*). The same region was then imaged for 30 min following the first generalized seizure, which began 45–75 min after injection. To study the longitudinal effects of status epilepticus in a separate cohort, a 15 min baseline recording was taken from two separate regions of the cranial window (55–80 µm depth). Kainate was then administered (19 mg/kg, i.p.) and seizures were visually monitored. After 1 hr, booster injections (7.5 mg/kg, i.p.) were administered every 30 min to animals not displaying a generalized seizure until such time as mice displayed at least eight generalized seizures. Chronic window mice meeting these inclusion criteria were then re-imaged in the same two locations 1, 2, 3, 7, 10, and 14 days after status epilepticus.

### Two-photon imaging: chemogenetic studies (*Figure 6*)

To validate the DREADD approach and determine CNO dose responses, we used WT C57BL/6J mice co-injected with AAV9.CaMKII.GCaMP6s and either AAV5.CaMKIIa.Gi-mCherry, or AAV5.CaMKIIa.Gq-mCherry. A cohort of mice injected with AAV9.CaMKII.GCaMP6s alone was employed for a CNO control study. To study the effects of CaMKIIa-DREADD effects on microglial calcium, we injected the Gq or Gi virus into GCaMP6s;Cx3Cr1$^{CreER}$ mice. All experiments were performed in chronic window animals (minimum of 4 weeks after window surgery and 7 weeks after virus injection). Somatic changes in neuronal calcium were studied in layer II/III (150–170 µm), while microglia were studied in layer I (55–75 µm depth). As a positive control for microglial studies, imaging was only performed in a region with high mCherry expression (visualized with a 740 nm excitation wavelength), and this region remained consistent across all trials. A trial consisted of a 15 min acclimation period to head restraint, 15 min baseline imaging session, removal and injection (saline or 1.25, 2.5, or 5.0 mg/kg CNO, i.p.), 15 min re-acclimation period under head restraint, then three 15 min post-injection imaging blocks. Post-injection changes in calcium activity were compared to the daily baseline period. CNO trials were separated by at least 48 hr, with lower doses administered first.

### Calcium image processing and ROI selection

T-series of neuronal and microglial calcium activity were motion corrected in ImageJ using the template matching plugin with subpixel registration. An average intensity image of morphology was then generated for ROI selection. For neuronal somata, ROIs were manually drawn with the circle tool for all neuronal somata detected in layer II/III, which appeared as a light ring with a dark circular nucleus. For microglia, two approaches to ROI selection were used. In both approaches, an average

intensity projection was created from the T-series. Notably, the low-intensity but constitutive eYFP signal of microglia is preferentially captured in the average intensity projection, providing an image of cell morphology that is not strongly influenced by overall calcium activity. In the first approach, used to characterize microdomain activity in *Figures 1* and *2*, somata and processes were manually segmented using the 'freeform' tool (see *Figure 2—figure supplement 1* for examples). Based upon their size and intensity, somata could be identified without ambiguity. Process segmentation followed three main rules: 1) the process had to be associated with a soma in the field of view, 2) all visible primary branches were segmented, and 3) any secondary branches emanating from a primary branch was segmented if greater than 15 pixels in length (~10 μm). This approach is referred to as 'manual segmentation' in the text and the resulting process ROIs are termed 'processes' in figures. In the second approach, semi-automated microglial process detection was utilized, as in *Figures 3–6*, when an intervention was introduced that could result in a hypothesized change in calcium activity. In this approach, termed 'threshold-based' ROI selection, the following steps occurred: 1) the unambiguous microglial somata were manually segmented by the freeform tool; 2) after acquiring their intensity values for ΔF/F calculations, these brighter somata were masked in the average intensity image to better threshold processes; 3) process territories were then identified in a semi-automated fashion using the threshold tool at a user-defined threshold level; 4) any process area larger than 50 pixels was included in the ROI manager; 5) these ROIs were then used to obtain intensity values. The threshold-based ROIs generated through these methods are termed 'process territories' as they often represent a primary process with multiple secondary and tertiary branches.

## Calcium activity analysis

Using the ROIs generated through the above methods, the multi-measure tool was used to obtain mean intensity values, which were converted into ΔF/F values using Excel. To determine the baseline fluorescence, we created a 100 s moving window average and then found the lower 25$^{th}$ quartile value across all frames. In certain cases of high activity, such as seizures during status epilepticus or spreading microglial calcium waves, the baseline was determined by finding the lower 25$^{th}$ quartile value in a user-selected, 200-frame period of non-seizure or non-wave activity. A calcium transient was considered to occur at a ΔF/F threshold over three times greater than the standard deviation of the baseline. The signal area represents the sum of any ΔF/F value above this threshold across all frames. A soma or process was considered active if it had a signal area of $\geq$5 ΔF/F•s.

## Process area and Sholl analyses

In experiments where motility changes were evaluated in real time (i.e. during the single-day experiments), we determined changes in overall process area (*Figures 3*, *4* and *6*). Using average intensity projections of the eYFP signal (full 600 or 900 frames), microglial somata were masked and a uniform threshold was applied to determine the process area. In longitudinal experiments (*Figures 2* and *5*), morphological complexity was determined using the Sholl analysis plugin of ImageJ (*Ferreira et al., 2014*).

## Motility analysis and assessment of extension or retraction

To determine process motility behavior, an average intensity image was generated from the first and last 100 frames of the T-series. From these images, a two-color overlay image was created (red: first; green: last). This image was used to initially highlight processes that underwent retraction (red color), extension (green color), or remained stable (red/green merge; see *Figure 3—figure supplement 1* for example). ROIs were manually created around each process then multiple steps were taken before final inclusion. First, all processes selected needed to be a minimum size for inclusion (>9 μm$^2$). Second, each process was visually inspected in the T-series for extension, retraction, or stability. Finally, processes needed to display at least a 50% area enlargement to be considered an extension (e.g. 10 μm$^2$ to 15 μm$^2$) or 50% area decrease to be considered a retraction (e.g. 10 μm$^2$ to 5 μm$^2$). Stable processes had less than a 50% change in area in either direction. After confirmation, these ROIs were also used to generate ΔF/F calcium traces to determine signal area and associated calcium activity based on each motility characteristic (*Figures 3I*, *5G*, *6K and L*). A process was considered to be associated with calcium activity if it had a signal area of >5 ΔF/F•s.

To determine temporal correlations between process extension/retraction and calcium events (*Figure 5H*), processes were selected as previously described using a manual ROI. We determined the start and end time of an extension/retraction event by using the threshold tool to create a frame-by-frame assessment of process area. The start of extension was defined as the first frame in a series of frames where process area became sequentially larger, while the end of extension was defined by a plateau in process area. The start of retraction was defined as the first frame in a series in which the process area began to reduce, while the end of retraction was defined by a plateau at a minimum value. We determined the temporal relationship between this extension/retraction start time and calcium events by determining the closest calcium event before and after the start time (if present, see *Figure 5—figure supplement 2* for examples). A calcium event needed to have a peak $\Delta F/F$ value of 0.5 to be considered, and only the nearest event before or after was considered. The latency between the peak of these calcium events relative to the start of extension/retraction was calculated for all extending/retracting processes in the *Figure 5* analysis and then binned by processes with an <30 s,<60 s, or <120 s latency between calcium event and extension/retraction start.

## General statistical analysis

A pilot study was first performed in acute window animals (N = 5) to determine the effect size and variability that isoflurane and acute kainate exposure had on microglial calcium activity. Based upon variability derived from these preliminary data, a power analysis was performed and an appropriately sized cohort was determined at a 0.05 alpha and 0.70 power level.

Two major effects were biologically replicated across two cohorts. We first observed the effects of kainate and isoflurane on microglial calcium activity in a pilot cohort (N = 5) using acute window animals. We then studied these effects in a separate cohort using chronic window animals (chronic window findings are reported in the present study after *Figures 1* and *2*). The effects of kainate and isoflurane were consistent in both cohorts and contexts. In addition, DREADD-based changes in both neuronal calcium activity and microglial calcium activity were technically replicated in the same respective cohorts after a 1-week delay. These replicates showed similar trends. Findings from the initial test are reported in the present study (not pooled or averaged). No outliers were removed from the data. The only grounds for animal study exclusion were based upon technical guidelines: two acute window animals were excluded due to post-operative cranial bleeding; approximately 30% of cranial window animals did not develop a sufficiently clear chronic window for adequate imaging and were excluded; two GCaMP6s;Cx3Cr1$^{CreER}$ mice in DREADD studies (one transfected with Gq virus and one transfected with Gi virus) were excluded due to a rare chronic window bleed or the loss of a headcap, respectively.

Prism (GraphPad, version 8) was used for group analyses. In most experiments, statistical significance was determined using a one- or two-way ANOVA design with Dunnett post-hoc comparison to the values obtained in the baseline period. Differences in the proportion of active microdomains were assessed using a Fisher's exact test. All data are expressed as the mean ± SEM. Statistical details are provided in the figures and figure legend.

## Acknowledgements

We thank Dr. Vanda Lennon (Mayo Clinic) for early insights on project direction and Yoga Varatharajah (Mayo Clinic) for assistance with code implementation. ADU is supported by an NIH F32 grant (NS114040). L-JW is supported by the Mayo Foundation and NIH RO1 grants (NS088627, NS110825, NS110949, and NS112144).

# Additional information

## Funding

| Funder | Grant reference number | Author |
| --- | --- | --- |
| National Institute of Neurological Disorders and Stroke | NS114040 | Anthony D Umpierre |
| National Institute of Neurological Disorders and Stroke | NS112144 | Gregory Worrell Long-Jun Wu |

| National Institute of Neurolo-gical Disorders and Stroke | NS088627 | Long-Jun Wu |
| National Institute of Neurolo-gical Disorders and Stroke | NS110825 | Long-Jun Wu |
| National Institute of Neurolo-gical Disorders and Stroke | NS110949 | Long-Jun Wu |

The funders had no role in study design, data collection and interpretation, or the decision to submit the work for publication.

## Author contributions

Anthony D Umpierre, Conceptualization, Data curation, Formal analysis, Funding acquisition, Validation, Visualization, Methodology, Writing - original draft, Writing - review and editing; Lauren L Bystrom, Data curation, Visualization, Methodology, Writing - review and editing; Yanlu Ying, Data curation, Writing - review and editing; Yong U Liu, Conceptualization, Methodology, Writing - review and editing; Gregory Worrell, Resources, Formal analysis, Methodology, Writing - review and editing; Long-Jun Wu, Conceptualization, Resources, Supervision, Funding acquisition, Writing - original draft, Project administration, Writing - review and editing

## Author ORCIDs

Anthony D Umpierre (iD) https://orcid.org/0000-0002-1470-8881
Long-Jun Wu (iD) https://orcid.org/0000-0001-8019-3380

## Ethics

Animal experimentation: All experimental procedures were approved by the Mayo Clinic's Institutional Animal Care and Use Committee (IACUC, protocol #2731-17) and were conducted in accordance with the NIH Guide for the Care and Use of Laboratory Animals.

## Decision letter and Author response

Decision letter https://doi.org/10.7554/eLife.56502.sa1
Author response https://doi.org/10.7554/eLife.56502.sa2

# Additional files

## Supplementary files

• Transparent reporting form

## Data availability

All data generated or analysed during this study are included in the manuscript and supporting files.

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
