## [Decision Letter]

Thank you for submitting your article "Microglial calcium signaling is attuned to neuronal activity" for consideration by *eLife*. Your article has been reviewed by two peer reviewers, and the evaluation has been overseen by a Reviewing Editor and Richard Aldrich as the Senior Editor. The reviewers have opted to remain anonymous.

The reviewers have discussed the reviews with one another and the Reviewing Editor has drafted this decision to help you prepare a revised submission.

As the editors have judged that your manuscript is of interest, but as described below that additional experiments are required before it is published, we would like to draw your attention to changes in our revision policy that we have made in response to COVID-19 (https://elifesciences.org/articles/57162). First, because many researchers have temporarily lost access to the labs, we will give authors as much time as they need to submit revised manuscripts. We are also offering, if you choose, to post the manuscript to bioRxiv (if it is not already there) along with this decision letter and a formal designation that the manuscript is 'in revision at *eLife*'. Please let us know if you would like to pursue this option. (If your work is more suitable for medRxiv, you will need to post the preprint yourself, as the mechanisms for us to do so are still in development.)

Summary:

This interesting and innovative study used in vivo two photon imaging to assess microglial calcium dynamics in the upper layers of the cerebral cortex in awake mice. The studies used a range of pharmacological and genetic methods to induce widespread changes in neuronal activity and correlated these to microglial calcium levels, process dynamics and cell size. The general conclusions are important – that microglia exhibit calcium dynamics that change with a substantial delay after altering neuronal activity, and that both increasing and decreasing neuronal activity leads to an increase in calcium and process extension.

Essential revisions:

1) The authors used CX3CR1-CreER-ires-EYFP mice crossed with cre-dependent GCaMP6s and lck-GCaMP6f calcium indicators, and thus performed simultaneous imaging of EGFP and EYFP. Apparently, there was no attempt to split these signals – due to their spectral overlap, it would be near impossible. As a result, the fluorescence changes reflect a combined GCaMP/EYFP signal. This needs to be clearly stated at the outset. As a result, the GCaMP signal to noise is reduced, so many smaller GCaMP fluctuations may have been missed. In the CX3CR1-CreER-ires-EYFP mice, the EYFP signal tends to be strongest in microglial somas – will this then preferentially impact the ability to detect subtle changes in GFP signal within microglial somas? This issue should be discussed. If the authors have data from animals expressing GCaMP only, it would be helpful to include some of these data for comparison. A second caveat with this approach is that all fluorescence fluctuations are interpreted as changes in intracellular calcium. Given the small size of these processes and their dynamic nature, there is concern that movements could also contribute to the fluctuations observed. This issue also needs to be discussed. Large scale coordination of events such as that shown in Figure 1E (Eii) raise concerns about such potential artefacts.

2) It is surprising that the authors did not use automated detection algorithms to identify and quantify microglia calcium transients and define microglial movements. A number of such programs have been developed and are readily available. The method of ROI categorization needs to be described more thoroughly and higher magnification images shown to validate the approach. They should show an example image that indicates a typical soma ROI and typical process ROIs with corresponding ΔF/F traces for those ROIs. (Figure 2C shows one cell with arrows but doesn't show the actual ROIs with some kind of outline). Without this information, there is confusion about what the ΔF/F traces represent. In Figures 1 and 2 – are these soma and individual process ROIs from a single representative cell or multiple process ROIs from different cells across one field of view? In Figures 3 and 4 where "process territory" ROIs are used – do the ΔF/F traces represent calcium signals in microglial processes in the entire field of view from different animals? Or process territories from different, individual microglial cells? What is the justification for examining individual process ROIs in Figures 1 and 2 but whole process territory ROIs in Figures 3 and 4? In Figures 5 and 6 the ΔF/F traces are labeled with "Processes" so it's unclear if individual ROIs on distinct regions of processes were used as in Figures 1 and 2 or if process territories were used as in Figures 3 and 4.

3) In trying to address the question about what are the basic patterns of calcium activity in microglial cells, the authors miss an opportunity to provide additional, useful information using the images they have already acquired. Are the cells that show soma calcium elevations the same group of cells that show process elevations? Or can you sometimes see cells that have only soma calcium elevations or only process calcium elevations? Of the cells that do show calcium elevations, are they scattered throughout the field of view or do they tend to be spatially clustered close to one another? What about these same questions following isoflurane, kainate, or DREADD manipulations? Also – if the mice are being imaged while on the NeuroTar air-lifted platform as described in the Materials and methods, can the authors comment about any relationship between microglial calcium elevations and ambulation or grooming?

4) One of the most interesting aspects of the work relates to linking process dynamics to changes in intracellular calcium. However, this phenomenon and quantification of this relationship needs to be illustrated better. For example, in Figure 3, the microglia are shown at such low magnification that it is not possible to visualize their processes. Indeed, many of the images are quite small and make it hard to judge how effectively the authors are able to visualize and analyze individual microglial processes (Figure 1B, Ei-iii, Figure 2C, D and G, Figure 5D, Figure 6J)In each such example, the authors need to provide several examples of individual processes and a time series of images showing the fluorescence changes that accompany the morphological changes. In trying to address questions about the relationship between microglial calcium elevations and process motility, the authors should do more to demonstrate how tightly associated (or not) are calcium elevations and process extensions. In Figure 3H, they indicate that 48% of process extensions occur in the absence of detectable calcium elevation. This would seem to indicate that these two processes are somewhat independent of one another. However, is this percentage derived from cells that they are certain express GCaMP because they've seen somatic or other process calcium elevations at some point during imaging? Or does this 48% include process extensions of cells that may not even express GCaMP? This same information (percentage of process extension/retraction/stable events that are associated with calcium elevations) should also be presented in the kainate (Figure 5) and DREADD (Figure 6) experiments. Figure 5F is potentially the most useful in showing whether there is a tight temporal relationship between process extension and calcium elevations. But we don't know if a similar temporal relationship is present in all processes that have both calcium elevations and extension or if there are also examples of processes that show extension and calcium elevations that are not temporally aligned. Can the authors show a similar panel to 5F for the isoflurane and DREADD experiments? For all instances of process extension accompanied by calcium elevations, can they quantify in some way the degree to which process extension and calcium elevations are temporally aligned and present this information?

5) In trying to address questions about the relationship between microglial calcium elevations and neuronal activity, the authors should similarly provide additional analyses of how closely aligned in time are the two phenomena. In the isofluorane and DREADD experiments (Figure 3 and Figure 6), the 10-15 min long bins used for quantification make it challenging to understand how closely aligned in time are changes neuronal activity and changes in microglial calcium signaling. We appreciate that using smaller time bins may be challenging, but in the absence of more detailed information about the onset of detectable changes in neuronal activity and how this relates to the onset of detectable changes in microglial calcium signals, statements that "microglia are attuned to changes in neuronal network activity" are not well supported. Perhaps microglia are tuned to changes in astrocyte state and this is modified by shifts in neuronal network activity.

More detailed information about the temporal relationship between changes in neuronal activity and microglial calcium activity could help with speculating about mechanisms involved. Were the Ca^2+^ transients in microglial processes only observed when neuronal activity specifically changed from normal levels to enhanced/silent levels? Or, were Ca^2+^ transients observed when neuronal activity changed more generally between different levels, e.g. low to normal, high to normal, high to low? We believe that it would be sufficient to explore this question in an acute brain slice preparation where it might be easier to manipulate neuronal activity by employing receptor/channel reagents. At the very least, this issue should be discussed in more detail.

The authors investigated the long-term relationship between Ca^2+^ transients in microglia and neuronal activity (Figure 5), but they did not check the neuronal activity in these experiments. For how long does the increase in neuronal activity last after kainate administration? Would this increase in neuronal activity be sustained and constant or more sporadic in nature? Could the change in neuronal activity in fact be more temporary itself but yet still trigger a sustained, long-lasting increase in Ca^2+^ transients in microglia? To address these questions, it is necessary to monitor neuronal activity throughout the entire time course following kainate administration, e.g. by EEG or neuronal Ca^2+^ imaging etc.

6) We need some indication about what the recombination efficiency is when giving tamoxifen in the diet as described. Otherwise, when presenting the percentage of microglial somas that exhibit calcium transients, we don't know if this is the percentage of GCaMP+ microglia that show spontaneous calcium elevations or if those cells that are "inactive" include cells that may not even express GCaMP.

7) The Discussion is quite brief and does not include any comments on either the nature of the delay in microglial responses or the function of the dynamic changes in process extension in relation to the changes in neuronal activity. I recommend that this section be extended. In their discussion of possible mechanisms regulating microglial calcium signaling, can the authors also comment on whether these calcium signals are likely to be solely driven by influx of calcium through plasma membrane receptors versus supplemented or driven by release of calcium from intracellular stores? Can they tease out any difference in the dynamics or magnitude of microglial calcium signaling following network inhibition and network excitation that would support the idea of different mechanisms driving microglial calcium signals in each context?

---

## [Author Response]

Essential revisions:1) The authors used CX3CR1-CreER-ires-EYFP mice crossed with cre-dependent GCaMP6s and lck-GCaMP6f calcium indicators, and thus performed simultaneous imaging of EGFP and EYFP. Apparently, there was no attempt to split these signals – due to their spectral overlap, it would be near impossible. As a result, the fluorescence changes reflect a combined GCaMP/EYFP signal. This needs to be clearly stated at the outset.

We thank the reviewers for the great comments and we have now clearly discussed the point.The initial studies did split the eYFP and GCaMP (‘GFP’) signals with a ‘GFP/YFP’ filter set (500/15 and 537/26). As shown theoretically with a spectral viewing program (Author response image 1) and confirmed empirically, splitting the signal significantly attenuates GCaMP detection in the GFP channel (19% of total) and does not prevent spectral overlap (30% of the GCaMP signal collected in the YFP channel). On the other hand, maintaining both signals in one channel without attenuation (520/15 filter) preferentially detected GCaMP emission (estimated 45% of total potential emission, not shown) over eYFP emission (26% of total potential emission, not shown). The filter information has been added to the Materials and methods section. Additionally, we recently recorded laser power at the objective instead of along the laser path and updated those values in text to give a more accurate evaluation of the laser power at the cranial window. We have also added a statement about the combined GCaMP/eYFP signal in the text both immediately in the Results (subsection “Spontaneous microglial calcium activity reported in the Lck-GCaMP6f and cytosolic GCaMP6s mouse”) and in the Discussion (subsection “The importance of microdomains in understanding glial calcium activity”).

**Author response image 1. respfig1:** Estimated collection efficiency of GCaMP6 and eYFP with a “GFP/YFP” filter set. Our initial approach to split these close spectra resulted in stronger GCaMP fluorescence in the YFP channel than the GFP channel and resulted in significant signal attenuation. Note that GCaMP6s was not an available fluorophore option, so GCaMP6f bound to calcium was used instead.

As a result, the GCaMP signal to noise is reduced, so many smaller GCaMP fluctuations may have been missed. In the CX3CR1-CreER-ires-EYFP mice, the EYFP signal tends to be strongest in microglial somas – will this then preferentially impact the ability to detect subtle changes in GFP signal within microglial somas? This issue should be discussed. If the authors have data from animals expressing GCaMP only, it would be helpful to include some of these data for comparison.

Yes. This does leave open the possibility for smaller fluctuations to be missed due to an artificially higher baseline (the F_0_ in our DF/F_0_). The potential under-reporting, particularly of smaller events has been discussed in the text (subsection “The importance of microdomains in understanding glial calcium activity”). Here we also concede that there is a potential bias for events in the soma to be particularly missed, but these events would need to be rapid and low amplitude. On the other hand, we also draw parallels between our work and other in vivo microglial calcium studies reaching similar conclusions about the rarity of somatic microglial calcium events (Eichhoff et al., 2011; Pozner et al., 2015).

Unfortunately, our mouse colony has historically kept the CX3CR1-CreER-IRES-eYFP line, not the CX3CR1-CreER line. Therefore, we do not have data from a mouse expressing GCaMP6s in microglia without the eYFP signal. However, we consider this a very important point and will be acquiring other microglia CreER lines soon to evaluate in future experiments.

A second caveat with this approach is that all fluorescence fluctuations are interpreted as changes in intracellular calcium. Given the small size of these processes and their dynamic nature, there is concern that movements could also contribute to the fluctuations observed. This issue also needs to be discussed. Large scale coordination of events such as that shown in Figure 1E (Eii) raise concerns about such potential artefacts.

This is a key point. If we understand your question, we might restate it as ‘any rapid movement in the x/y/z plane could result in a region of fluorescence moving in and out of the ROI, resulting in what erroneously appears as a calcium change.’ We took the example of Figure 1E further to analyze this point and suggest the following: If there is a rapid shift in the focal plane in any direction, one would expect both a negative deflection (fluorescence moves out of the ROI, leading to a decrease in intensity reading) followed by a positive deflection (fluorescence moves back into the ROI, increasing the intensity reading). After subpixel registration, we do not see major decreases in fluorescence intensity, only positive increases (Author response image 2). Further, we utilized the ImageJ registration plugin (Align Slices in Stack) for Figure 1E to tell us how much correction (in pixel distance) occurred from one frame to another. We plotted this pixel displacement as a proxy to quick shifts in the focal plane, presumably due to animal movements. This is plotted against the DF/F traces gathered in Figure 1E. We can see no clear correlation between shifts in the imaging frame and calcium events. The synchronized and coordinated calcium signals are hypothesized to reflect neuronal activity changes sensed by surrounding microglial processes. In the Discussion (subsection “Spontaneous microglial calcium activity and the utility of different GCaMP variants and preparations”), we discuss a potential hypothesis and future direction to study this phenomenon.

**Author response image 2. respfig2:** Exploring movements and potential erroneous calcium detection. Adapted from Figure 1E, we do not see that frame-by-frame shifts (black line, pixel displacement detected by registration plugin) correlates with increases in calcium activity.

2) It is surprising that the authors did not use automated detection algorithms to identify and quantify microglia calcium transients and define microglial movements. A number of such programs have been developed and are readily available.

We thank the reviewers for this important point of discussion. Multiple automated programs were tested over the past year for calcium detection.

1) Inscopix data processing software: We first attempted to use the PCA/ICA package embedded in the Inscopix data processing software to automatically detect microglial process events after masking the soma region across all frames. We found this program often would overestimate or underestimate process size or create redundancies.

2) Romano et al.’s computational toolbox:We also tried the Romano et al., 2017 program published in PLOS Computational Biology. Multiple processing steps in this toolbox were helpful, but ultimately, the user-guided parameters available for ROI detection were not ideal for detecting the complex shape of microglial processes (Romano et al., 2017).

3) GECI Quant:Dr. Bal Khakh’s GECI Quant program, an ImageJ-based detection program is ultimately very similar to our overall approach for process territory detection. GECI Quant first conducts frame-by-frame background subtraction, then has the user hand-segment somata, then has the user draw a bounding box around a territory of processes, which are detected at a user-defined threshold level. These steps generate threshold-based ROIs used in DF/F calculations. Our processing steps in finding process territories are very similar: (1) we created an average intensity projection, (2) hand-drew ROIs around somata, (3) used multi-measure to obtain soma intensity values for DF/F calculations, (4) masked these somata to better threshold processes, (5) threshold-selected processes with the analyze particles tool and added them to the ROI manager to create process territory ROIs, then (6) used multi-measure to obtain intensity values for these process territories used in DF/F calculations. We could have used GECI Quant in our analysis, except the data tabulation and program work-through is much slower than the simplified macros we created to preprocess data and obtain intensity values. Ultimately, we would argue that our process territory approach is reasonably similar to other programs that are considered semi-automated ROI detection approaches (Srinivasan et al., 2015).

The method of ROI categorization needs to be described more thoroughly and higher magnification images shown to validate the approach. They should show an example image that indicates a typical soma ROI and typical process ROIs with corresponding ΔF/F traces for those ROIs. (Figure 2C shows one cell with arrows but doesn't show the actual ROIs with some kind of outline).

Yes, the Materials and methods section has been updated to more thoroughly explain the two different approaches to ROI selection (subsection “Calcium image processing and ROI selection”). We have created Figure 2—figure supplement 1 to address this question. In this figure, we expand upon Figure 2A to show 4 different microglia cells with highlighted soma and process ROIs alongside DF/F traces. This gives a better sense of the cell-to-cell variability and how hand-drawn ROIs were created. Additional examples of higher-magnification images were added to all figures.

Without this information, there is confusion about what the ΔF/F traces represent. In Figures 1 and 2 – are these soma and individual process ROIs from a single representative cell or multiple process ROIs from different cells across one field of view? In Figures 3 and 4 where "process territory" ROIs are used – do the ΔF/F traces represent calcium signals in microglial processes in the entire field of view from different animals? Or process territories from different, individual microglial cells?

In Figures 1 and 2, ∆F/F traces are from the soma and processes of a single, representative cell. This has been updated in the figure legend.

In Figure 3D and Figure 4D, the microglial DF/F traces were created by the threshold-based ROI method and represent traces attributable to one representative cell, not a random selection of ROIs. For Figure 3E (heat map), the microglial process territories shown are all of the threshold-based process ROIs detected in different microglial cells from a single, representative animal across one experiment. The text of the figure legend has been updated to provide better clarification.

What is the justification for examining individual process ROIs in Figures 1 and 2 but whole process territory ROIs in Figures 3 and 4?

In brief, the justification for using the two methods relates to perception over biased ROI selection. Automated segmentation of individual microglial processes is difficult due to their small size and tortuosity. Across multiple approaches, we found that no method to identify individual processes was as accurate as manual segmentation. In Figures 1 and 2, our purpose was to provide readers with characterization of process and soma calcium activity. We did not feel that this characterization presented strong cause for concern regarding bias, so we wanted to use the best approach we could employ. In Figures 3 and 4, we are testing whether an intervention like isoflurane or kainate alters microglial calcium activity. Herein, we wanted to avoid any concern that potential user bias in ROI selection could influence the results. For this reason, we used a threshold-based ROI selection technique to reduce user influence in ROI selection.

In Figures 5 and 6 the ΔF/F traces are labeled with "Processes" so it's unclear if individual ROIs on distinct regions of processes were used as in Figures 1 and 2 or if process territories were used as in Figures 3 and 4.

Yes, these ROIs were generated through the threshold-based ROI approach and should be termed ‘process territories.’ This has been stated clearly in the figures and in the figure legend.

3) In trying to address the question about what are the basic patterns of calcium activity in microglial cells, the authors miss an opportunity to provide additional, useful information using the images they have already acquired. Are the cells that show soma calcium elevations the same group of cells that show process elevations? Or can you sometimes see cells that have only soma calcium elevations or only process calcium elevations?

Thank you for bringing up this question. We have performed this analysis and added the results as Figure 2—figure supplement 1B. Microglia with somatic activity are not more likely to have process activity. Our results showed that about 25% of all microglia surveyed have soma and process calcium activity, while approximately 50% of microglia have process calcium activity without somatic calcium activity. Specifically, cells with only somatic calcium elevations are quite rare (<5%). On the other hand, about 50% of microglia have process activity without somatic activity, so this represents the predominant pattern of activity.

Of the cells that do show calcium elevations, are they scattered throughout the field of view or do they tend to be spatially clustered close to one another? What about these same questions following isoflurane, kainate, or DREADD manipulations?

In regards to spontaneous calcium activity in both acute and chronic window preparations, microglial cells with calcium activity are scattered throughout the field of view. With isoflurane application, at the peak of activity (10-30 minutes), increased calcium activity can be seen throughout the field of view, but some areas do represent ‘hot spots’ of dramatically increased activity. With kainate application during status epilepticus, or Gi/Gq activation, we observe similar results, where active cells are scattered throughout the field of view.

Also – if the mice are being imaged while on the NeuroTar air-lifted platform as described in the Materials and methods, can the authors comment about any relationship between microglial calcium elevations and ambulation or grooming?

This is a great question. Our current NeuroTar platform was an original version without magnetic location tracking. We will soon update our NeuroTar platform to have location tracking ability and plan to perform this experiment in the motor cortex. Similar to grooming, we found that deflecting whiskers evokes microglial calcium activity in the barrel cortex. However, these results are preliminary and we intend to utilize this approach to uncover the molecular mechanism behind this signal in the future. However, these results are preliminary and we intend to utilize this approach to uncover the molecular mechanism behind this signal in the future.

4) One of the most interesting aspects of the work relates to linking process dynamics to changes in intracellular calcium. However, this phenomenon and quantification of this relationship needs to be illustrated better. For example, in Figure 3, the microglia are shown at such low magnification that it is not possible to visualize their processes. Indeed, many of the images are quite small and make it hard to judge how effectively the authors are able to visualize and analyze individual microglial processes (Figure 1B, Ei-iii, Figure 2C, D and G, Figure 5D, Figure 6J) In each such example, the authors need to provide several examples of individual processes and a time series of images showing the fluorescence changes that accompany the morphological changes.

We thank the reviewers for the great question. Here are the additional analysis to address the reviewers’ concerns: For Figure 1B we added a zoomed-in image of a microglial cell with processes for both the Lck-GCaMP6f and cyto-GCaMP6s mouse. For Figure 1E, we did not perform process-based analyses and instead adopted a grid-based approach to describe the diffuse signal that largely originates in more parenchymal spaces. For Figure 2 panels, we created Figure 2—figure supplement 1 to demonstrate multiple microglia and the creation of manual ROIs to study their motility. For Figure 5D, we added a zoomed-in image of an individual cell. For Figure 6J, we created Figure 6—figure supplement 1 to highlight microglia processes.

We initially created time series images of processes (see Figure 3—figure supplement 1A and B) and attempted to inter-relate these images to expanded DF/F traces. However, we realized that doing this for all the requested figures with multiple examples for extension, retraction, or stable processes would take up a lot of figure space. We would kindly request to instead show such dynamics with videos, such as Figure 5—video 2 for the Figure 5 dataset. Additionally, motility and calcium can be viewed in Figure 3—video 1 for the isoflurane study and Figure 6—video 1 for the DREADD studies. However, we agree that better visualization of these phenomenon would improve the manuscript and we created multiple figure supplements to show a two-color image of extending, retracing, or stable processes and their calcium traces (Figure 3—figure supplement 1, Figure 5—figure supplement 2, Figure 6—figure supplement 1).

In trying to address questions about the relationship between microglial calcium elevations and process motility, the authors should do more to demonstrate how tightly associated (or not) are calcium elevations and process extensions. In Figure 3H, they indicate that 48% of process extensions occur in the absence of detectable calcium elevation. This would seem to indicate that these two processes are somewhat independent of one another. However, is this percentage derived from cells that they are certain express GCaMP because they've seen somatic or other process calcium elevations at some point during imaging? Or does this 48% include process extensions of cells that may not even express GCaMP? This same information (percentage of process extension/retraction/stable events that are associated with calcium elevations) should also be presented in the kainate (Figure 5) and DREADD (Figure 6) experiments.

Thanks for the great questions. This percentage has been derived from all cells/processes we could detect in the field of view. We did not include/exclude cells based upon whether any part of the cell had GCaMP activity. However, as explained in response to Question 6, we find that recombination is pretty high in our studies (at least 85%), so non-expression of GCaMP would not likely explain the ~50% of process extensions occurring without calcium activity. To address the reviewers’ concerns, we have performed the additional analysis: The percentage of process extension/retraction/stable events associated with calcium elevations has been added for the kainate figure (Figure 5G) in place of the cumulative distribution plot. This has also been added for the Gi and Gq DREADD datasets (Figure 6K and L).

Figure 5F is potentially the most useful in showing whether there is a tight temporal relationship between process extension and calcium elevations. But we don't know if a similar temporal relationship is present in all processes that have both calcium elevations and extension or if there are also examples of processes that show extension and calcium elevations that are not temporally aligned. Can the authors show a similar panel to 5F for the isoflurane and DREADD experiments? For all instances of process extension accompanied by calcium elevations, can they quantify in some way the degree to which process extension and calcium elevations are temporally aligned and present this information?

In addition to Figure 5F data, we have found a cleaner way to display this information with less figure space by highlighting the start and stop of extension/retraction overlaid with the DF/F trace (now Figure 5—figure supplement 2). This visualization approach was used for the isoflurane (Figure 3—figure supplement 1) and DREADD experiments (Figure 6—figure supplement 1).

We performed temporal analysis for the data one day after kainate status epilepticus (Figure 5H and Figure 5—figure supplement 2). In this dataset, we find that 51% of extending processes have a calcium event occurring within 1 minute of extension start, compared to 18% in retracting processes.

5) In trying to address questions about the relationship between microglial calcium elevations and neuronal activity, the authors should similarly provide additional analyses of how closely aligned in time are the two phenomena. In the isofluorane and DREADD experiments (Figure 3 and Figure 6), the 10-15 min long bins used for quantification make it challenging to understand how closely aligned in time are changes neuronal activity and changes in microglial calcium signaling. We appreciate that using smaller time bins may be challenging, but in the absence of more detailed information about the onset of detectable changes in neuronal activity and how this relates to the onset of detectable changes in microglial calcium signals, statements that "microglia are attuned to changes in neuronal network activity" are not well supported. Perhaps microglia are tuned to changes in astrocyte state and this is modified by shifts in neuronal network activity.

This is a great suggestion. For the isoflurane experiments, we created Figure 3H to analyze process territory calcium signal area using one-minute bins. These values were then compared using a one-sample t-test to the average baseline period value to determine when significant changes in calcium signal area are first detected. We find significant process calcium activity above baseline is detected within 6 minutes after isoflurane exposure.

However, we began this analysis for the DREADD experiments and found it difficult to draw any sounder conclusions. In the isoflurane experiments, we know: (1) that there was a clear start time for neuronal silencing (within 1 minute after induction, Figure 3C); (2) that neuronal activity was consistently silenced across all animals (~90%, Figure 3B-D); (3) that these results should be generalizable between cohorts. On the other hand: (1) DREADD-based effects take time to develop (progressive increase or decrease in activity, Figure 6D), while gas anesthetic action is near instantaneous; (2) variability in the time-of-peak effect exists between animals; (3) DREADD effects in the neuronal study cohort are broadly generalizable to the microglial study cohort, but probably cannot be used to draw firm minute-by-minute conclusions. After performing the minute-by-minute analysis, we still find that microglial calcium activity peaks approximately 15 minutes after the peak change in neuronal activity (Author response image 3).

**Author response image 3. respfig3:** Minute-by-Minute relationship between microglial calcium activity and neuronal activity changes. An example of breaking down neuronal calcium activity (top) into 3 min bins correlated with changes in microglial calcium activity. Neuronal calcium changes are significant as soon as 15 min after injection. Significant microglial changes are first detected 33 min after injection, suggesting a ~15 min latency between the two. These significant changes are sustained for the remainder of the recording.

The latency in the microglial response is a very important point for discussion and interpretation in regards to attunement to neuronal activity. We have dedicated a major section of our Discussion to this consideration (subsection “The relationship between neuronal activity changes, process extension, and process calcium signaling”). In addition, we agree that our results do not demonstrate a direct relationship between neuronal activity and microglial process signaling (i.e. real-time neurotransmitter release is the impetus for the observed microglial calcium signals). We have added this key point to the Discussion (subsection “The importance of microdomains in understanding glial calcium activity”). However, we also think it is important to consider that in DREADD-based experiments, neuronal activity changes are the start of a biological process that results in microglial calcium signaling increases.

More detailed information about the temporal relationship between changes in neuronal activity and microglial calcium activity could help with speculating about mechanisms involved. Were the Ca^2+^ transients in microglial processes only observed when neuronal activity specifically changed from normal levels to enhanced/silent levels? Or, were Ca^2+^ transients observed when neuronal activity changed more generally between different levels, e.g. low to normal, high to normal, high to low? We believe that it would be sufficient to explore this question in an acute brain slice preparation where it might be easier to manipulate neuronal activity by employing receptor/channel reagents. At the very least, this issue should be discussed in more detail.

We thank the reviewers for this insightful point of consideration. This is indeed a current area of exploration in brain slices, representing a large undertaking to determine the mechanism(s) of microglial calcium signaling. Therefore, we would like to develop future projects that move from slice to an in vivo context exploring identified mechanisms. We discussed the question of state changes in relation to our current model (Discussion, subsection “The relationship between neuronal activity changes, process extension, and process calcium signaling”). Based upon our current work, we would hypothesize that state changes back to baseline may not trigger notable calcium elevations in microglia. These state changes are anticipated to involve more retraction than extension as microglia return to a less extended/surveilling state. Our work predicts that state changes engaging process retraction would not result in elevated calcium activity in microglia.

The authors investigated the long-term relationship between Ca^2+^ transients in microglia and neuronal activity (Figure 5), but they did not check the neuronal activity in these experiments. For how long does the increase in neuronal activity last after kainate administration? Would this increase in neuronal activity be sustained and constant or more sporadic in nature? Could the change in neuronal activity in fact be more temporary itself but yet still trigger a sustained, long-lasting increase in Ca^2+^ transients in microglia? To address these questions, it is necessary to monitor neuronal activity throughout the entire time course following kainate administration, e.g. by EEG or neuronal Ca^2+^ imaging etc.

While we did not perform these experiments using neuronal calcium imaging or EEG, the first author has previously investigated the systemic kainate mouse model extensively using 24/7 video EEG (Umpierre et al., 2018, 2016). In his previously published studies, spectral analysis of cortical EEG from the parietal cortex indicates that neuronal activity broadly falls into three periods:

In the first 3 days after kainate status epilepticus, neuronal activity remains elevated. This could be considered evidence that more sustained elevations in neuronal activity occur in this period. EEG traces in this period commonly show interictal spiking as well as some sub-clinical (focal-cortical) seizure activity that does not secondarily generalize. These events could be considered more sporadic manifestations of coordinated neuronal activity increases.

Between 3-7 days after kainate status epilepticus, increased electrographic power begins to decrease and subclinical seizures are no longer observed; however, interictal spiking can persist, but normally at a reduced frequency.

Between 7-14 days after kainate status epilepticus, electrographic power is not noticeably increased. Interictal spiking is variable: persisting in some animals, gone in others, or manifesting sporadically over 24-72hr periods in others.

Spontaneous, generalized seizures are rare in the systemic kainate model and are not seen until 14 days after status epilepticus at earliest. Therefore, it is unlikely that spontaneous, epileptic seizures factored into the data. For multiple logistical reasons, it is very difficult to implant an EEG electrode in a mouse with a Neurotar headplate and retain this surgical assembly for many weeks, which is why we did not attempt to perform simultaneous, near-24/7 video EEG alongside longitudinal calcium imaging.

6) We need some indication about what the recombination efficiency is when giving tamoxifen in the diet as described. Otherwise, when presenting the percentage of microglial somas that exhibit calcium transients, we don't know if this is the percentage of GCaMP+ microglia that show spontaneous calcium elevations or if those cells that are "inactive" include cells that may not even express GCaMP.

This is an important point that we initially intended to study; however, when we tried to design the experiment, we realized it could be quite difficult. In brief, the sequence similarity between eYFP and GCaMP (a circularly permutated GFP molecule) represented a strong concern for interpreting IHC data if antibodies detect a conserved epitope. If so, we could not tell all CX3CR1+ cells with constitutive eYFP expression from recombined cells with additional GCaMP expression.

Alternatively, we provide some evidence of recombination efficiency. We looked at the longitudinal study of microglial calcium activity following status epilepticus in chronic window animals (N=5). Our closest proxy to a positive control (similar to ATP application in culture or slice experiments) would be spreading wave calcium activity, where a large event propagates throughout the entire area of imaging (N=3/5 animals), or during the day of highest activity in the absence a wave (N=2/5 animals). To determine if a cell viably expressed GCaMP, we counted whether any part of a microglial cell exhibited calcium activity during these videos across all cells clearly identifiable in the area. After identifying cells with no activity, we also searched longitudinally through the dataset looking at other imaging days to determine if these non-active cells had an event on another day. Using this approach, we found that between 85-100% of microglia had a calcium response somewhere within the cell over multiple days (Author response image 4; mean ± SEM: 89 ± 6%). This is decent evidence from a cohort of male and female mice that our recombination is likely strong.

**Author response image 4. respfig4:** Longitudinal survey of active microglia in the kainate cohort. (Left) The survey method was used to observe any clear calcium event within any part of a cell (green outline) or confirm no observable event in the 15-minute recording (red outline). This example image can also be watched as Figure 5—video 1: Spreading Calcium Wave. In recordings from another day, the inactive cell (#2, red outline) did show somatic calcium activity, suggesting the cell did have GCaMP expression when studied longitudinally. (Right) Percentage of microglia exhibiting calcium activity within one 300 x 300 µm region surveyed over six days (performed across N=5 mice).

7) The Discussion is quite brief and does not include any comments on either the nature of the delay in microglial responses or the function of the dynamic changes in process extension in relation to the changes in neuronal activity. I recommend that this section be extended. In their discussion of possible mechanisms regulating microglial calcium signaling, can the authors also comment on whether these calcium signals are likely to be solely driven by influx of calcium through plasma membrane receptors versus supplemented or driven by release of calcium from intracellular stores? Can they tease out any difference in the dynamics or magnitude of microglial calcium signaling following network inhibition and network excitation that would support the idea of different mechanisms driving microglial calcium signals in each context?

Our manuscript was originally developed as a short report, leading to a shortened Discussion. We have expanded our Discussion to include the following topics: (1) discussion on the eYFP signal and how it impacts the study, (2) the nature of the delay in microglial responses and how it relates to motility and neuronal changes, (3) ionotropic versus metabotropic signaling mechanisms based upon previous in vivo work by others addressing this distinction, and (4) future directions and hypotheses regarding the Lck-GCaMP6f signal. Potential mechanisms are also discussed, which we are currently pursuing in brain slice.

In the Discussion, we mention other research (Eichhoff et al., 2011; Pozner et al., 2015) showing that both calcium influx pathways and intracellular calcium stores can be activated in vivoin microglia, but intracellular calcium stores appear to be more important for the injury responses studied in prior research. Based on the calcium signature observed in cytosolic-GCaMP6s microglia (including event size, spread, kinetics, and amplitude), we would have to posit that most of what we observe is consistent with intracellular store calcium. However, the calcium activity observed in the Lck-GCaMP6f mouse leaves open the possibility that this very rapid, small amplitude and diffuse calcium signature could potentially be calcium influx. We intend to test selective ionotropic and metabotropic receptor pharmacology in slice to discern the mechanism(s) for calcium elevations in microglia, but cannot give a definitive answer at this time.

Unfortunately, we cannot tease out any difference in the dynamics or magnitude of microglia calcium signaling after network inhibition or excitation. In DREADD-based experiments, approximately equal increases or decreases in neuronal calcium activity (i.e. a 31% decrease versus a 42% increase) results in very similar increases in microglial calcium signaling (64% versus 68% peak calcium signaling elevations) and have similar latencies. Thus, we do not detect major differences in the timing or magnitude of microglial calcium changes in response to reasonably equivalent shifts towards hyperactivity or hypoactivity.

References:

Romano SA, Pérez-Schuster V, Jouary A, Candeo A, Boulanger-Weill J, Sumbre G. 2017. A Computational Toolbox and Step-by-Step Tutorial for the Analysis of Neuronal Population Dynamics in Calcium Imaging Data. bioRxiv 103879+. doi:10.1101/103879

Umpierre AD, West PJ, White JA, Wilcox KS. 2018. Conditional Knockout of mGluR5 from Astrocytes during Epilepsy Development Impairs High-Frequency Glutamate Uptake. J Neurosci 39:1148–18. doi:10.1523/JNEUROSCI.1148-18.2018